# Optimal Pricing for Data-Augmented AutoML Marketplaces

**Minbiao Han** [* 1]  **Siyuan Xia** [* 1]  **Jonathan Light** [2]  **Raul Castro Fernandez** [1]  **Haifeng Xu** [1]  **Sainyam Galhotra** [3]

## Abstract

Data markets promise to unlock data value by matching data suppliers with ML consumers. However, market design involves addressing intricate challenges, including data pricing, fairness, and robustness. We propose a pragmatic data-augmented AutoML market that seamlessly integrates with existing cloud-based AutoML platforms, such as Google's Vertex AI. Unlike standard AutoML solutions, our design automatically augments buyer-submitted training data with valuable external datasets, pricing the resulting models based on their measurable performance improvements rather than computational costs as the status quo. Our key innovation is a pricing mechanism grounded in the instrumental value—the marginal model quality improvement—of externally sourced data. This approach bypasses direct dataset pricing complexities and accommodates diverse buyer valuations through menu-based options, thus providing an economically sustainable framework for monetizing external data.

## 1. Introduction

Many organizations that would benefit from machine learning lack access to the necessary training data, which is costly to obtain (Laffont & Martimort, 2009; Najafabadi et al., 2015; Miotto et al., 2018). At the same time, other organizations sit on top of large volumes of data that could be helpful to the former. Data markets that match the supply of datasets with demand would unlock tremendous value. Buyers would submit their ML tasks, the market would automatically find training data suitable to that task and train a model that satisfies the buyers' demands.

Some recent proposals such as (Roth, 2018; Milgrom & Tadelis, 2018; Epasto et al., 2018; Hu & Chen, 2018) have

---

*Equal contribution [1]University of Chicago, USA [2]Rensselaer Polytechnic Institute, USA [3]Cornell University, USA. Correspondence to: Siyuan Xia <stevenxia@uchicago.edu>.

*Proceedings of the 43$^{rd}$ International Conference on Machine Learning*, Seoul, South Korea. PMLR 306, 2026. Copyright 2026 by the author(s).

explored mechanisms for pricing data, focusing on maximizing sellers' revenue. Yet, these approaches face significant challenges for real-world deployment: (i) they typically price data based on the raw dataset itself, rather than on the actual utility it provides through a trained machine learning model; and (ii) they often lack transparency in how prices are determined. Moreover, none of these proposals considers data augmentation to improve model accuracy. To build a fair and sustainable data marketplace, we need a principled methodology that prices datasets based on their demonstrated utility to buyers while accommodating a diverse willingness to pay. In this work, we address the following question: What is the best way to price datasets that align incentives and maximize overall market performance?

We take a pragmatic approach to designing the market: rather than contributing a clean-slate model or mechanism, we design a data-augmented AutoML market that functions as a drop-in replacement of existing cloud-based AutoML platforms, such as Google's Vertex AI (Cloud, 2023) and Amazon's SageMaker (Joshi, 2020). In AutoML platforms, buyers send training data to the platform. The platform performs the model search and returns a high-performing one, and buyers pay for the search cost. The market we design uses similar interfaces, but differs fundamentally in two aspects: (1) our platform (which implements our market design) *augments* buyer's initial training data with additional features on the platform; (2) buyers primarily pay for the resulting model (and, implicitly, for the identified data) according to its estimated quality improvement, rather than the cost of computation or model search. This shift in emphasis toward pricing the model quality reduces buyer participation risk and enables a new market-based mechanism for monetizing external data.

A central challenge is developing a mechanism to price discovered data and models. We define the instrumental value of data-augmented autoML (Castro Fernandez, 2025; Ai et al., 2025) as the marginal improvement in model quality attributable to both model development and external data. Our market presents buyers with a menu of options, each associated with a different value described by model quality and corresponding price. This design allows buyers to select what is appropriate for them. There are a few key properties of our pricing mechanism: i) Pricing based on instrumental value reduces individual buyer's risk of paying

much computing cost to get a unuseful model. This encourages more participation. ii) Since we do not price individual datasets directly, properties such as volume or completeness are implicitly accounted for, removing the need for difficult-to-specify modeling assumptions. iii) The menu includes multiple options, targeting buyers with different willingness-to-pay (i.e., types).

To price instrumental value of data-augmented autoML, our market design needs to identify useful external data and models for a buyer's task. In our design (Section 3), buyers submit a training dataset, and the market system searches a large underlying data pool to enhance performance. We build upon existing techniques for the data and model discovery component, whose output will be served as a basis for our pricing algorithm.

Overall, our solution is API-compatible with AutoML platforms, and additionally integrates external data into the learning process and prices outcomes based on their measured improvement, paving the way for the monetization of external data. Our evaluation shows that our pricing algorithm effectively compensates participants by maximizing generated revenue compared to other baselines, and our market is effective in leveraging external data to find high-quality models for a wide range of buyer tasks.

## 2. Related Work

**Pricing Problem.** The contract design literature (Myerson, 1982; Hart, 2016) studies the problem where a platform sets a contract of payments to maximize the expected revenue. Contract theory has played a significant role in economics (Grossman & Hart, 1992; Smith, 2004; Laffont & Martimort, 2009), and there has been a growing interest in the computer science community regarding the problem of contract design driven by the increasing prevalence of contract-based markets in web apps (Alon et al., 2021; Castiglioni et al., 2022; Gan et al., 2022). These markets have significant economic value, with data and computation playing a central role. In contrast to traditional contract design challenges, our design is on crafting contracts, specifically pricing curves in a data-augmented autoML market setting. Online pricing against strategic buyers is also widely studied in the economic literature (Dütting et al., 2011; Celis et al., 2011; Dawkins et al., 2021; Drutsa, 2017). Prior research concentrated on analyzing various aspects of equilibria, with optimization efforts dedicated towards maximizing data owner's revenue. Our work takes a broader perspective by addressing the optimization of both buyer utility and overall market welfare.

**ML and Data Markets.** Today's commercial online data marketplaces, such as Snowflake's, AWS's, sell raw data (Snowflake, 2023; Amazon, 2023); we concentrate on selling the instrumental value of data, identifying relevant data by its impact on the buyers' tasks, avoiding the main challenge of helping buyers decide how raw data helps their purpose. On the other hand, existing AutoML markets misses on the opportunity to augment buyer-submitted training data with external data to train higher-quality models. Lastly, many ML markets have been proposed in previous work (Agarwal et al., 2019; Chen et al., 2019; Sun et al., 2022; Liu et al., 2021; Song et al., 2021). While they have considered selling the instrumental value of data, they assume a model is built over all data in the market, and study how to sell versioned models to buyers with different preferences. Yet, in a market with a large data collection, only a small subset may be relevant for any given task. Our market takes the data-model search problem into design, which is directly reflected in our pricing mechanism.

## 3. Data-Augmented ML Market Architecture

We now present our design of a data-augmented AutoML market. At a high level, *buyers* bring their tasks and preliminary training data to a *platform* (the market). The platform has a collection of datasets, possibly contributed by various data owners. The platform will conduct both data augmentation search and model search via automated machine learning (autoML), where an augmentation refers to additional features from external datasets in the data collection that are joined with the buyers' training data. Compared to the status quo of autoML market such as Vertex AI and Sage Maker, the core novelty in our design lies on (a) the transparency about the evolution of model performance based on which each buyer can decide when the data and model search should stop to avoid paying cost for not-so-useful searches; and (b) a pricing mechanism that is *primarily* based on revealed model performance rather than the total amount of consumed computing regardless of what model performance the buyer receives. This design gives more control to buyers and better quality assurance, hence could potentially attract many more buyers. Next, we illustrate our design in detail.

**Basic Setup.** We study how to design an autoML market that helps users find both good data and models.

Each buyer arrives at the platform with a machine learning task. The platform then searches for a model $C$, as well as additional features to enhance the original training dataset $D$. Let $q(C \leftarrow D)$ denote the *model quality* of the obtained model (e.g., accuracy on some test data). The model quality will be calculated on buyer's provided validation data. While buyers submitting a validation set may raise questions on strategic behaviours (e.g, curating an easier evaluation set), we note that doing so is risky for the buyer: strong performance on a biased set need not transfer to the actual workload, and any loss from that mismatch is borne by the buyer. Manipulating the evaluation data in a way that both lowers price and still guarantees deployment performance requires a high degree of expertise from the buyer, which is

atypical of the targeted users of our AutoML market.

To let buyers reduce risk and control stopping time, our designed market will reveal to buyers the current model quality $q_t \in Q$ at pre-specified time periods $t = 1, 2, \cdots, T$ (e.g., every 5 minutes). We assume $q_t$ has finite resolution and is drawn from a discrete set $Q$ of all possible (rounded) qualities. Notably, $q_{t+1}$ may be smaller than $q_t$ since it is the *latest* discovered model quality, which may become worse due to searching over worse datasets during the search. Revealing all discovered model qualities so far, instead of just the best quality, is a tailored design choice since we would like to provide users with a variety of choices, because some users may prefer lower-quality model at a lower price.

Since many ML algorithms are based on gradient descent (Haji & Abdulazeez, 2021; Zhang, 2019), a model's future performance is often independent of previous parameters given its current performance. This motivates our modeling of the evolution of model performance $q_t$ as a Markov chain. This model is particularly natural for modern optimization-driven model training like gradient descent, where updates are driven by the immediate surroundings of current parameters (Ruder, 2016). We thus denote the accurate evaluation of the data discovery procedure as a Markov chain $\langle Q, T, \{\boldsymbol{P}_t\}_{t \in [T]} \rangle$, in which $\boldsymbol{P}_t(q_t | q_{t-1})$ is the transition probability from previous state $q_{t-1}$ to the current state $q_t$. It is important to allow $\boldsymbol{P}_t$ to be time-dependent because the probability of transitioning from model quality $q$ to $q'$ typically differs a lot at different times (e.g., the later, the less likely to have a big increase). We assume this transition matrix $\boldsymbol{P}$ is public information, which may be learned from the platform history and disclosed to users as a "suggested guidance" (a similar example is Google's ad exchange revealing data to advertisers about how to bid (Google, 2023)).

**Modeling buyer preferences.** Following standard assumptions in mechanism design (Maskin & Riley, 1984; Myerson, 1979; 1982), we assume there is a population of buyers who are interested in this autoML service. Each buyer is modeled by a private *type* $\theta \in \Theta$. Formally, each $\theta$ determines a *vector* $v^\theta(\cdot) \in \mathbb{R}_+^Q$, in which $v^\theta(q)$ describes her values for model quality $q \in Q$. While the buyer knows his type $\theta$, the platform only knows a prior distribution $\mu$ over the buyers. We will start by assuming that the platform knows the distribution $\mu$, then discuss how to relax this assumption using a learning-theoretic approach.

**Data Discovery Process and the Interaction protocol.** Our mechanism relies on a data and model discovery procedure that supplies model qualities in real-time. During this discovery process, the user sees the qualities of all discovered models. At any time, they may choose to stop the process, and *purchase any discovered model, or leave the market*. Our designed mechanism for pricing data-augmented ML models is represented by a price curve (PC) $x \in \mathbb{R}_+^Q$, which prescribes a price $x(q)$ for each possible model quality $q \in Q$, and is posted in advance to the potential buyers.

**Buyer Payment and the Optimal Pricing Problem.** The payment of any buyer consists of two components: (1) *the data discovery cost* $c\tau$ in which $\tau \in [T]$ is the user's stopping time and $c$ is the *true* computation cost per unit of time; (2) *payment* $x(q)$ per inference if the buyer chooses to purchase a data-augmented ML model with quality $q$ for inference. Hence, buyer $\theta$'s (quasi-linear) utility is $u^\theta(q, x, c) = v^\theta(q) - x(q) - c\tau$, where $v^\theta(q)$ denotes the buyer's value of model quality $q$, $x(q)$ denotes the price charged for this model quality, and $c$ denotes the search cost for each unit of time. The payment of data discovery costs is mandatory to sustain and compensate the system operation's costs. However, we expect this cost to be very low comparing to buyer's valuation of a good model. (For example, a powerful EC2 instance with 8 vCPU costs 0.27/hr for compute. (Amazon, 2025)) The payment of $x(q)$ per inference is voluntary and represents what our platform truly sells. The platform's pricing problem is hence, given a prior distribution $\mu$ regarding buyer types $\Theta = \{\theta_1, \cdots, \theta_n\}$, and the performance dynamics $\{\boldsymbol{P}_t\}_{t \in [T]}$ of model qualities, finding the price curve $\{x(q)\}_{q \in Q}$ that maximizes the expected market revenue.

**Key Challenges to Address.** For the designed market above to work, we face three major technical challenges: (1) how to find the optimal pricing mechanism efficiently, (2) how can the market place obtain the prior distribution $\mu$ about buyers, (3) to have a truly functional market, we also need a discovery process to effectively search for models and data, and reveal model qualities in real time. Next, we address these challenges in order, all through computational approaches, including an implementation of our designed market for evaluation and measuring its effectiveness.

## 4. Computing the Optimal Pricing Mechanism

In this section, we focus on the first key challenge: designing a pricing mechanism to compute the optimal price that aligns incentives and maximizes overall market performance.

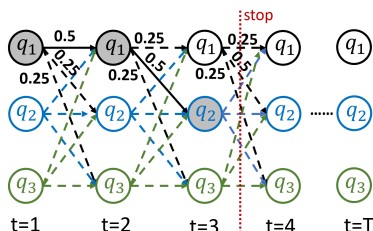

*Figure 1.* A Markov chain model where the buyer makes a decision to continue or stop at every node.

**Buyer's best response as optimal stopping.** Since the buyer knows the price and search cost before starting dis-

covery, she faces an optimal stopping problem in our market. Specifically, the buyer's search for model quality is a Pandora's Box problem, a widely studied online optimization problem to balance the cost and benefit of search (Weitzman, 1978; Boodaghians et al., 2020; Ding et al., 2023). Here, an agent is presented with various alternatives modeled by a set of boxes $B = \{b_1, \cdots, b_n\}$, where each box $b_i$ needs cost $c_i$ to be opened, and has a random payoff $v_i$ whose distribution is known. The agent's goal is to find a strategy that adaptively decides whether to stop or continue the search after opening each box. Similarly, in our problem, the buyer also needs to make the decision to continue or stop at every time step $t \in [T]$. A major difference between our problem and Pandora's box problem is that they assume all boxes' payoff distributions are *independent*, while in our problem, the buyer's random payoff of the next time step (i.e., box) depends on the state of the current one. This dependence is modeled through the Markov chain $\{\boldsymbol{P}_t\}_{t \in [T]}$.

For any buyer type $\theta$, let us consider the case where the buyer faces a price curve $x \in \mathbb{R}_+^Q$. At time step $t$, suppose the *maximal surplus* model quality the buyer has seen is denoted as $q = \max_{t'=1}^{t} v^\theta(q_{t'}) - x(q_{t'}) - c \cdot t'$, the realized quality at time step $t$ is denoted as $q_t$, we denote $\tau(q, q_t, t)$ as the (random) stopping time conditioned on the events $\tau \geq t$. Given any $\tau$, we can define its corresponding expected future reward starting in state $(q, q_t, t)$ for the buyer as $\phi^\tau(q, q_t, t) = \mathbb{E}[\max(v^\theta(q) - x(q) - c \cdot t, \max_{t'=t+1}^{\tau(q, q_t, t)} v^\theta(Q_{t'}) - x(Q_{t'}) - c \cdot t')]$, where $Q_t$ denotes the random variable and $\max_{t'=t+1}^{\tau(q, q_t, t)}$ denotes the best model quality the buyer can find until the stopping time $\tau(q, q_t, t)$. We define $\Phi(q, q_t, t) = \max_\tau \phi^\tau(q, q_t, t)$ to represent the expected optimal future reward the buyer can achieve in state $(q, q_t, t)$ under optimal policy $\tau^* = \arg\max_\tau \phi^\tau(q, q_t, t)$. Thus, the buyer's goal is to compute a $\tau^*$ that maximizes $\phi^\tau(q_0, q_0, 1)$, i.e., the buyer's expected future reward starting at the initial quality $q_0$ at the starting time $t = 1$. We show the buyer can efficiently compute the optimal policy via well-designed dynamic programming (DP).

**Proposition 4.1.** *The optimal buyer policy can be computed via DP in $O(Q^2 T)$ time.*

Our DP algorithm constructs a DP table $\Phi$ with shape $Q \times Q \times T$. Each entry $\Phi(q_1, q_2, t)$ in the table records the buyer's optimal expected future reward the buyer can achieve at state $(q_1, q_2, t)$, i.e., at time step $t$, the model quality with the maximal surplus the buyer has seen is $q_1$, while the realized model quality at time step $t$ is $q_2$. Since the underlying process is a known MDP, the buyer's future expectations depend only on the current model quality $q_2$. We can then calculate buyer's optimal stopping policy $\tau^*$ by comparing the difference between buyer's utility if they stop right now vs. the expected future utility of continuing.

*Proof.* We propose the following Algorithm 1 to compute

the optimal buyer decisions with respect to a given price curve $x \in \mathbb{R}_+^Q$. The dynamic program algorithm 1 has a DP

---

**Algorithm 1** OptimalBuyerDecisions

**Input:** Price curve $x \in \mathbb{R}_+^Q$; **Output:** Optimal policy $\tau^*$
1: Initialize DP table $\Phi(Q, Q, T)$, where each element $\Phi(q_1, q_2, t) = \max_\tau \phi^\tau(q_1, q_2, t)$ (see Section 4).
2: Initialize policy table $\tau^*$ of size $Q \times T$ where $\tau^*(q, t) = 0/1$ represent buyer stops/continues at $q$ at time $t$.
3: **for** $q_1, q_2 \in Q$ **do**
4:     Compute the base-case utility at $t = T$, i.e., $\Phi(q_1, q_2, T) = v^\theta(q_1) - x(q_1) - c \cdot T$, since $T$ is the last round the buyer can only stop.
5:     Set $\tau^*(q_2, T) = 0$.
6: **end for**
7: **for** $t = T - 1, \cdots, 1$ **do**
8:     Compute $\Phi(q_1, q_2, t)$ by a DP solved in decreasing order of $t$, i.e.,

$$\Phi(q_1, q_2, t) = \max \Big( v^\theta(q_1) - x(q_1) - c \cdot t,$$
$$\mathbb{E}_{\boldsymbol{P}(q|q_2)} \big[\Phi(\underset{\{q_1, q\}}{\arg\max}(v^\theta(q_1) - x(q_1), v^\theta(q) - x(q)), q, t + 1)\big]\Big). \quad (1)$$

9:     **if** $v^\theta(q_1) - x(q_1) - c \cdot t$ is larger in (1) **then**
10:         Set $\tau^*(q_2, T) = 0$ (i.e., buyer should stop)
11:     **else**
12:         Set $\tau^*(q_2, T) = 1$ (i.e., buyer should continue)
13:     **end if**
14: **end for**
15: **return** $\tau^*$

---

table with $Q \times Q \times T$ states, where filling each state takes constant time. As a result, the total time of the algorithm is $O(Q^2 T)$, proving the proposition. $\square$

**Finding optimal pricing curve from empirical data.** We now show how to approximate the optimal pricing curve using the formulation of a mixed integer linear program.

Given a common prior distribution $\mu$ over the buyer population and sampled quality trajectories from the empirical data discovery process. Specifically, the Markov chain model $\{\boldsymbol{P}_t\}_{t \in [T]}$ can be estimated through the realized quality trajectories set $S$, where $s \in S$ represents a sequence of model qualities discovered (i.e., $s = [q_1, \cdots, q_T]$). The following theorem guarantees the computation of a solution from a Mixed Integer Linear Programming (MILP) that provably approximates the optimal pricing curve under mild assumptions. The MILP can be solved efficiently by industry-standard solvers such as Gurobi (Gurobi Optimization, LLC, 2022) or Cplex (Cplex, 2009).

**Theorem 4.2.** *Suppose the discovery cost $c$ is negligible compared to the buyer's value. Then using $m = |\widehat{S}| = \frac{b^2 \ln 1/\delta}{2\epsilon^2}$ samples in S, the following MILP with variable $x$ computes a pricing curve that is an additively $\epsilon$-approximation to the optimal curve with probability at least $1 - 2\delta$. Here, $b$ is the maximum valuation from any buyer,*

*and $\epsilon > 0$ is the error term.*

$$\max \quad \sum_{\theta} \mu(\theta) \sum_{s \in S} \frac{1}{m} \sum_{q^s \in s} y(\theta, s, q) \qquad (2)$$

$$\text{s.t.} \quad 0 \leq a_{\theta, s} - \left[ v^{\theta}(q) - x(q) \right] \leq M\left(1 - z(\theta, s, q)\right),$$

$$\sum_{q \in s} z(\theta, s, q) = 1,$$

$$y(\theta, s, q) \leq x(q), \; y(\theta, s, q) \leq M z(\theta, s, q),$$

$$y(\theta, s, q) \geq x(q) - \left(1 - z(\theta, s, q)\right) M,$$

$$\boldsymbol{x} \geq 0, \; \boldsymbol{z} \in \{0, 1\}, \; \boldsymbol{y} \geq 0.$$

This theorem demonstrates that a near-optimal pricing curve can be learned purely from sample trajectories, backed by strong PAC-style guarantees. It highlights the approach's sample efficiency and robust generalization, key advantages of PAC learning, making it a practical method for data-driven pricing. A detailed analysis of the MILP can be found in Appendix A.3. We note that the decision variables $x(q)$ in the MILP are precisely the pricing curve itself, which maps every model quality $q$ to a price $x(q)$. The buyer's optimal stopping behavior is characterized by the decision variable z's in the MILP. The constraint about $z$'s in the MILP is capturing that $z$ has to represent an optimal buyer response. Program (2) is a bi-level optimization program that maximizes the platform's payoff, assuming that the buyers will optimally respond. We briefly discuss the core assumption of the above theorem, i.e., the assumption that the data discovery cost $c\tau$ is negligible compared to the model price and buyer valuations. Note that the discovery cost $c\tau$ in our pricing scheme is different from the payment in the existing AutoML market, which charges buyers $\alpha\tau$ for running on their cloud for $\tau$ time.[1] The $\alpha$ in their pricing model is the price per unit of computation, whereas $c$ in our scheme is the *true* computation cost. Generally, $\alpha >> c$.

**Assumptions on Market's Observations of the Trajectories.** In practice, larger providers (e.g., Vertex AI) observe many trajectories for similar workloads, giving them the ability to log trajectories from prior training runs across various tasks. Examples of different classes of tasks include demand forecasting for different months for retail stores like Walmart (Kunal Banerjee, 2022). Having access to such trajectories allows the markets to continually improve their estimations of the transition probabilities as input to the pricing MILP. We will also show later in Theorem 5.2 how to quantify loss as a function of estimation errors.

## 5. Relaxing Market's Prior through Learning

While assuming knowledge of the prior distribution over buyer types is standard in economic models, real-world markets often operate under uncertainty and incomplete

information. However, we expect the market to perform well even without access to the true prior. We now study how the seller can gradually learn the prior distribution $\mu^*$ through repeated interactions with buyers. We further quantify the impact of using an approximately optimal prior by bounding the resulting revenue loss.

Our goal is to learn the true distribution $\mu^*$ by iteratively updating our belief based on observed stopping times, beginning with a uniform prior $\mu^{(0)}$. In each round $t$, we observe a stopping time $y^{(t)}$ in response to the offered pricing curve $\boldsymbol{x}^{(t)}$. For any type $\theta_i \in \Theta$, we define $\mathbf{Pr}\left(y^{(t)} \mid \theta_i, \boldsymbol{x}^{(t)}\right)$ as the probability that a buyer of type $\theta_i$ stops at $y^{(t)}$ when presented with $\boldsymbol{x}^{(t)}$. To simplify notation, we fix the buyer type $\theta$ in the following discussion and omit it from the notation. Let $\mathbf{Pr}_t(q)$ be the probability that the buyer has not stopped before time $t$ and that the maximum observed utility has quality $q$. According to the buyer's optimal policy, computed via Algorithm 1, each time step $t \in [T]$ is associated with a reservation value $z_t$. This value represents the threshold at which the buyer is indifferent between continuing and stopping. That is, the buyer proceeds if the difference between the highest observed quality and its price is below $z_t$, and stops otherwise. The reservation value is the smallest solution to:
$$\mathbb{E}\left[ \left( \max\nolimits_{j=t}^{\tau^*(z_t, q_{t-1}, t)} X_j - z_t \right)_+ - c \cdot (\tau^*(z_t, q_{t-1}, t) - t)_+ \right] = 0,$$
where $X_j$ is the buyer utility distribution $v(q_j) - x(q_j)$ according to transition matrix $\boldsymbol{P}(\cdot | q_{t-1})$ for all $q_i \in Q$, and $\tau^*(z_t, q_{t-1}, t)$ is the optimal random stopping time given that the max buyer utility sampled in the past has been $z_t$ and the buyer has just passed $t - 1$, or nothing if $t = 1$.

---

**Algorithm 2** Bayesian Learning of Market Prior Knowledge

1: **Input:** Initial prior $\mu^{(0)}$, learning rates $\{\eta_t\}$, reservation thresholds $\{z_t\}$
2: **Phase 1: Compute stopping behavior**
3:     **for** each buyer type $\theta_i$ and round $t \in [T]$:
4:        Compute $\mathbf{Pr}(q, q_{t-1}, z_t)$ (see Eq. (3)), the probability that the buyer stops at round $t$, the max observed utility's quality is $q$, and the previous quality is $q_{t-1}$.
5:        Sum over all $(q, q_{t-1})$ to get the probability that type-$\theta_i$ stops at round $t$.
6: **Phase 2: Run Bayesian learning over $N$ trials**
7:     **for** trial $t = 1$ to $N$:
8:        Sample type $\theta^{(t)}$ and simulate stopping round $y^{(t)}$.
9:        For each type $\theta_i$, retrieve the precomputed probability that type-$\theta_i$ would stop at round $y^{(t)}$.
10:       Use Bayes' rule to update belief: $w^{(t)}(\theta_i) \propto$ (stopping probability) $\times$ (prior belief).
11:       Smooth the posterior into the new prior: $\mu^{(t+1)} = (1 - \eta_t)\mu^{(t)} + \eta_t w^{(t)}$.

---

Given the reservation values, we compute the joint prob-

---

[1] At the time of this paper's writing, Vertex AI has $\alpha = \$3.5$ per node hour for Google's AutoML service (Cloud, 2023).

ability $\mathbf{Pr}(q, q_{t-1}, z_t)$ that the buyer stops at time $t$, the maximum observed model quality is $q$, and the quality at the previous step is $q_{t-1}$ (for $t > 1$). This probability is given by $\mathbf{Pr}(q, q_{t-1}, z_t) = \mathbf{Pr}_t(q \mid q_{t-1}) \cdot \mathbb{1}_{q > z_t}$, where $\mathbf{Pr}_t(q \mid q_{t-1})$ denotes the probability that the maximum quality reaches value $q$ at time $t$, conditioned on having value $q_{t-1}$ at the previous time step. This probability is defined recursively as follows:

$$\mathbf{Pr}_t(q \mid q_{t-1}) = \begin{cases} \boldsymbol{P}_1(q), & \text{if } t = 1, \\ \boldsymbol{P}_t(q \mid q_{t-1}) \sum_{q' < q} \mathbf{Pr}_{t-1}(q') & \text{otherwise.} \\ \quad \cdot \mathbb{1}_{q' \leq z_{t-1}} \\ \quad + \mathbf{Pr}_{t-1}(q) \mathbb{1}_{q \leq z_{t-1}} \\ \quad \cdot \sum_{q' \leq q} \boldsymbol{P}_t(q' \mid q_{t-1}) \end{cases} \quad (3)$$

As a result, we can update the posterior belief about the buyer's type, denoted as $w^{(t)}$, by the Bayesian update rule as: $w^{(t)}(\theta_i) = \frac{\mathbf{Pr}\left(y^{(t)} \mid \theta_i, \boldsymbol{x}^{(t)}\right) \mu^{(t)}(\theta_i)}{\sum_{\theta_j} \mathbf{Pr}\left(y^{(t)} \mid \theta_j, \boldsymbol{x}^{(t)}\right) \mu^{(t)}(\theta_j)}$, and update $\mu^{(t+1)}$ as $\mu^{(t+1)} = (1 - \eta_t) \cdot \mu^{(t)} + \eta_t \cdot w^{(t)}$, where $\eta_t$ is the learning rate which, e.g., can be set to $\frac{1}{t+1}$. Details can be found in Algorithm 2. More results on how the learning rate choices affect convergence are provided in Section 6. The Bayesian update process is guaranteed to converge to the true distribution $\mu^*$ under standard posterior concentration results, assuming identifiability, i.e., no two distinct distributions $\mu_1$ and $\mu_2$ induce identical buyer responses under all seller learning strategies (Doob, 1949; Schwartz, 1965). Note that this assumption is without loss of generality, since if two buyer types are indistinguishable to the seller's learning algorithm, they can be treated as a single type.

While the Bayesian learning process above converges to true distribution, one concern is on the potential loss the market has to suffer due to having inaccurate estimate of the distribution. We now quantify this revenue loss, measured by the *Cost of Estimation Error* (CEE), and show that as long as the learned prior and transition matrices are sufficiently accurate, CEE remains small. Our learning-based approach hence is not only consistent but also robust in terms of revenue performance.

**Definition 5.1** (Cost of Estimation Error (CEE)). *Let $\mu^*$ be the true buyer distribution and $\boldsymbol{P}^* = \{\boldsymbol{P}_t^*\}_{t \in [T]}$ be true transition matrices. For any estimated prior $\mu$ and transition matrices $\boldsymbol{P}$, let $x^*[\mu, \boldsymbol{P}]$ denote the optimal pricing curve computed under $\mu$ and $\boldsymbol{P}$, and let $Rev_{\mu, \boldsymbol{P}}(x)$ denote the expected revenue of pricing curve $x$ under buyer distribution $\mu$ and transition matrices $\boldsymbol{P}$. The CEE is then defined as:* $\text{CEE}(\mu, \boldsymbol{P}) = \text{Rev}_{\mu^*, \boldsymbol{P}^*}\left(x^*[\mu^*, \boldsymbol{P}^*]\right) - \text{Rev}_{\mu^*, \boldsymbol{P}^*}\left(x^*[\mu, \boldsymbol{P}]\right)$. *Note that $\text{CEE}(\mu, \boldsymbol{P}) \geq 0$ holds for any $\mu$ and $\boldsymbol{P}$. We next show that CEE remains bounded when using an approximately optimal $\mu$ and $\boldsymbol{P}$.*

**Theorem 5.2** (Bounding Cost of Estimation Error). *Suppose learned $\mu$ satisfies $||\mu - \mu^*||_2 \leq \epsilon$ and transition matrices $\boldsymbol{P}$ satisfies $||\boldsymbol{P} - \boldsymbol{P}^*||_2 \leq \epsilon$, then $\text{CEE}(\mu, \boldsymbol{P}) = O(\epsilon)$.*

The formal proof is provided in Appendix B. This result generalizes our learning result in Theorem 4.2, by removing the assumption on negligible discovery cost.

## 6. Evaluation

We now present our experimental contribution to the third challenge: an implementation of the proposed data-augmented autoML market with an effective data-model discovery process. We then evaluate the effectiveness of our implementation along the following dimensions: its ability to find good augmentations-model pairs over a range of buyers' tasks, the effectiveness of the pricing scheme, and how well we can learn the prior distribution.

**An Implementation of the Market and Experimental Setup** Recall that our pricing mechanism requires a sequence of model model qualitys to create the menu of quality-price pairs. Yet, the augmentation-model discovery step is highly non-trivial, especially when the data collection is large. A brute-force approach that considers every combination of sets of augmentations $\mathbf{P}$ and ML models $\mathbf{M}$ gives the optimal solution, but it is intractable to train $O(2^{|\mathbf{P}|}|\mathbf{M}|)$ models where $|\mathbf{P}|$ is often in the order of millions. To implement the discovery engine for our evaluation, we generalize a well-known existing data discovery engine called *Metam* (Galhotra et al., 2023) that prunes the search space. Specifically, Metam leverages properties of the data (datasets that are similar often yield similar model qualitys on ML models) to prune irrelevant augmentations and prioritizes them in the order of their likelihood to benefit the downstream task. A straightforward adaptation of (Galhotra et al., 2023)'s technique would rank the different augmentations, train $\mathbf{M}$ model to test the top-ranked augmentation, then use the realized performance to update the rankings for later iterations. This approach can be shown to converge in $O(\log \mathbf{P})$ rounds, leading to $O(|\mathbf{M}| \times \log |\mathbf{P}|)$ model trainings, which is too slow to run.

To make our implementation practical, we developed an efficient mechanism to choose the best augmentation and model jointly. Specifically, we model each ML model as an arm and simulate a bandit-based learning problem to choose the best model. Note that our problem is similar to pure exploration framework (Chen et al., 2014), as the goal is to choose the best model and stop exploration after that.

Algorithm 3 presents the pseudocode of our augmentation-model discovery mechanism. It uses Exp3 (Auer et al., 2002) to choose the best arm, while searching for the augmentations. In each iteration, a model is chosen based on a probability distribution as specified in line 5, and trained

**Algorithm 3** AUGMENTATION-MODEL DISCOVERY ALGORITHM

**Input**: Training Data $D_{tr}, D_{val}$, List of Models $\mathbf{M}$, Augmentations $\mathbf{P}$

1: Initialize the solution set of augmentations $\mathbf{T} \leftarrow \phi$; Initialize $w(M) = 1, \forall M \in \mathbf{M}, \gamma = 0.1$
2: **WHILE** $i \leq$ STOPPINGCRITERION
3:   Choose a candidate augmentation $T_i \subseteq \mathbf{P}$, and set $\text{Pr}(M) = (1 - \gamma)\frac{w(M)}{\sum_{M \in \mathbf{M}} w(M)} + \frac{\gamma}{|\mathbf{M}|}$
4:   $M_i \leftarrow$ Sample a model $M$ according to the probability distribution $\text{Pr}$.
5:   Update $w(M_i) = w(M_i)e^{\gamma \widehat{r}(M)}/|\mathbf{M}|$, where $\widehat{r}(M) = m(M_i \leftarrow (D_{in} \bowtie T_i)) * 1.0/Pr(M_i)$
6:   $u(T_i) \leftarrow m(M_i \leftarrow (D_{in} \bowtie T_i))$
7: RETURN $arg \max_M m(M \leftarrow D_{in}, \widehat{T})$ FOR $\widehat{T} \leftarrow arg \max\{u(T), \forall T \in \mathbf{P}\}$

on the input dataset with augmentation $T_i$. $T_i$ is chosen according to a sampling-based approach from (Galhotra et al., 2023) that clusters the augmentations in $\mathbf{P}$ and greedily chooses the augmentation that achieves the maximum gain in the model quality. The model quality $m$ evaluated on the trained model is then used as reward to update $w(M)$. The main advantages of the Exp3 algorithm are that the adversarial bandit-based formulation helps with the changing input dataset in each iteration, and it has shown superior empirical performance in practice (Bubeck et al., 2009). The STOPPINGCRITERION implements the *anytime* property, letting buyers stop searching whenever they want. Notably, Algorithm 3 trains a single ML model in each iteration as opposed to $|\mathbf{M}|$ trainings by prior techniques, leading to convergence in $O(\log |\mathbf{P}|)$ iterations as opposed to $O(|\mathbf{M}| \times \log |\mathbf{P}|)$. We implement this as the discovery building block for our market, and will measure its effectiveness empirically in the following experiments. More details and convergence results of the discovery component are in Appendix.

**RQ1: How effective is our market in finding high-quality augmentations and models?** We implement the proposed market, including the pricing mechanism and the above augmentation-model discovery component, on a market with $69K$ datasets (with $\approx$ 30M columns and 3B rows) from the NYC open data. The market searches over these datasets to identify the best combination of augmentations and models (random forest, XGBoost, neural network, etc.) that help improve buyers' input tasks. We present the average results on a sample of 1000 classification and regression tasks. Examples include predicting the performance of different schools on standardized tests in NYC (2000 records; 6 features), and predicting the number of collisions using daily trip information (400 records; 4 features). All initial datasets contain $> 500$ records. We consider three competitive baselines to compare the effectiveness of our discovery algorithm. (i) DATA-ALL uses (Galhotra et al., 2023) to search for the best augmentations and trains all ML models

in each iteration. (ii) DATA-ALT fixes a high-efficiency, low-cost ML model (random forest classifier in this case) to test the chosen augmentation in each round and then trains an AutoML model (using the TPOT library) on the best augmentation every minute. (iii) AUTOML approach does not search over datasets, only models using TPOT. We represent our approach as DATA-BANDIT.

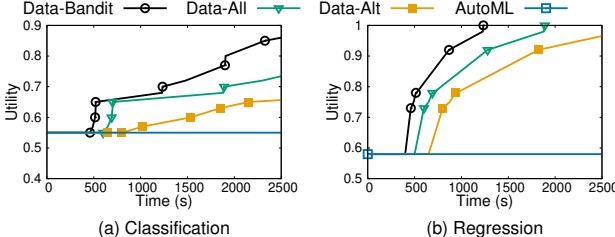

(a) Classification    (b) Regression

*Figure 2.* Comparison of discovery techniques on 1K tasks.

Figure 2 reports the average results on the 1000 different ML tasks. We observe that the bandit-based approach used by our discovery engine outperforms all baselines consistently across the considered tasks by achieving a higher utility for the buyers in a shorter amount of time. For classification tasks, the average utility found by our platform is higher than 0.85, whereas the second-highest benchmark is just above 0.7. AutoML does not search for data, and its average utility for classification is around 0.55. For regression tasks, our platform also performs better: fix any point on the x-axis (search time), and our platform achieves higher utility than all baselines. For most tasks, the number of augmentations identified is less than 10. Notice that a higher running time of the discovery algorithm translates to higher waiting times and computing costs for the buyer. Although the additional computation cost incurred by higher running time is small for buyers with high valuations, it becomes more significant for buyers with low valuations of tasks. Our approach can identify the best model consistently across different tasks, and is effective in generating a suite of options for the buyer to choose from based on their requirements.

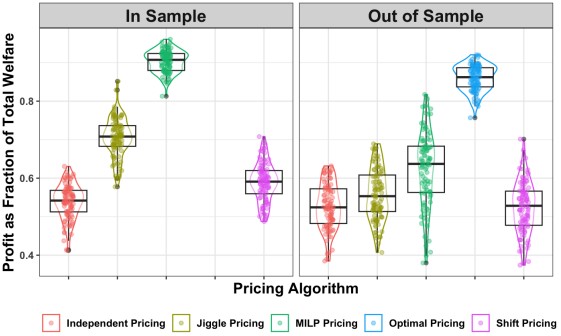

*Figure 3.* Profit benchmark for school data.

**RQ2: How do different pricing schemes perform?** We now study how much profit our pricing algorithms generate, as a fraction of the total possible welfare, for both in sample

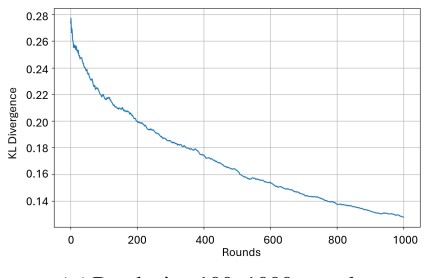

*(a)* Batch size 100, 1000 rounds.

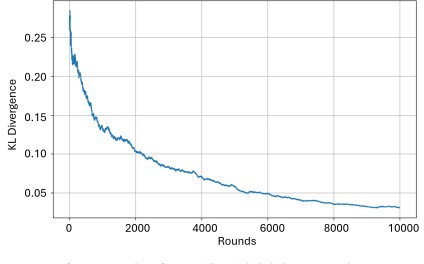

*(b)* Batch size 10, 10000 rounds.

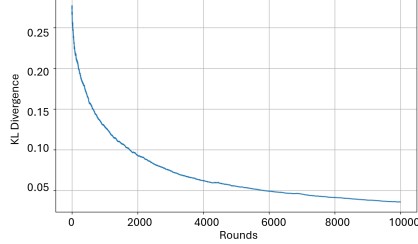

*(c)* Batch size 100, 10000 rounds.

*Figure 4.* Results under a randomly generated prior distribution $\mu^* \in \Delta^n$. The model quality has $|Q| = 10$, e.g., $Q = \{100\%, 99\%, \ldots, 91\%\}$. The number of buyer types is $n = 5$, time steps $T = 15$, and learning rate $\eta = 1/\sqrt{t}$. The $x$-axis shows learning rounds; the $y$-axis shows KL divergence between the learned and true distributions.

(IS) and out of sample (OOS) model quality trajectories. We train our pricing algorithms on the sample, produce pricing curves, and evaluate the total profit.

We generate the buyer valuations for both IS and OOS using the same random process and draw a new independent sample every time. In IS training, all model qualitys are available to the buyer, while in OOS, a random subset of model qualitys is drawn each time from real model qualitys that buyers might encounter using real data. We evaluate the model qualitys for different tasks to identify a plausible set of model qualitys and limit the buyer to only choosing from a randomly sampled limited pool of model qualitys OOS. This places a significant limitation on our pricing algorithm while OOS, which creates a good test of robustness. In addition to the nearly optimal pricing curve computed by our MILP formulation in Equation (2), we present three pricing baselines in increasing order of complexity and flexibility to fit to the sample data, which help demonstrate how bias and variance interact in the context of pricing. Furthermore, we also included the optimal pricing scheme for OOS if all model qualitys were available to the buyer. The three baselines are:

**Independent pricing scheme.** Given a prior distribution $\mu \in \Delta^{|\Theta|}$ over the buyer types, where each type $\theta$ has a value $v_q[\theta]$ for model quality $q$, we can consider the valuation $V_q$ as a random variable with $\mathbb{P}(V_q = v_q[\theta]) = \mu(\theta)$. The independent pricing scheme assumes that $V_q$ is independent of $V_{q'}$ for any $q \neq q'$, and computes the price for each model quality that maximizes the expected payment with respect to the prior distribution $\mu$: $x(q) = \arg\max_x \mathbb{E}_\theta[x \mathbb{1}_{v_q[\theta] \geq x}]$.

**Shift pricing scheme.** This scheme shifts the independent pricing scheme up or down by some discrete shift parameter $k \in \{-|\Theta|, -|\Theta| - 1, ..., 0, ..., |\Theta|\}$. Specifically, let $\theta_q^0$ be the $\theta$ chosen by the independent pricing scheme. Let $\theta_q^k$ be $\theta$ such that $v_q[\theta_q^k]$ is $k$-ranks higher or lower than $v_q[\theta_q^0]$ if we sort $\{v_q[\theta_q] | \theta \in \Theta\}$ from smallest to largest. Then the shift-$k$ pricing scheme sets the price $x(q) = v_q[\theta_q^k]$.

**Jiggle pricing scheme.** The jiggle pricing algorithm takes an initial selection of $\theta_q$ values and tests out small changes to them to see if it improves pricing. At each step, JIGGLE tests out two possible modifications to $\theta_q$: (1) find a $q$ that has a high probability of being chosen and shift the price up to $x(q) = v_q[\theta_q^1]$ so that we charge more for popular model qualitys (2) find a $q$ that has a low probability of being chosen and shift the price down to $x(q) = v(q)[\theta_q^{-1}]$ so that we charge less for undesired model qualitys.

Figure 3 displays the performance of our algorithms in the IS and OOS setting, respectively [2]. In both figures, the y-axis represents the normalized, expected profit as a fraction of the total welfare (i.e. how much total surplus the buyers can get if all prices are set to 0). The total welfare represents an upper bound on how well any mechanism can do. The higher a pricing scheme is, the more profit it is able to capture and the better it is. In the IS setting, the MILP outperforms our other algorithms, capturing around 85% of the total welfare, whereas all others capture less than 80%. In the adverse OOS setting, MILP also outperforms the other baselines, only trailing the optimal pricing scheme. It is still able to capture a majority of the welfare.

**RQ3: How to learn the prior distribution $\mu$?** We now present the results on learning the prior distribution. Specifically, we assume the buyer's types are distributed according to an underlying distribution $\mu^*$. We validate the learning method by running it over various distributions, including a distribution that's generated randomly, a uniform distribution, a slightly skewed distribution, a highly skewed distribution, and an extremely skewed distribution. We also vary the learning rate $\eta_t$ from $\frac{1}{t+1}$ to $\frac{1}{\sqrt{t}}$ and $\frac{1}{2}$, representing more conservation to aggressive learning rates. In addition, instead of updating the posterior distribution according to $\mu^{(t+1)} = (1 - \eta_t) \cdot \mu^{(t)} + \eta_t \cdot w^{(t)}$ with a single posterior distribution, we smooth the learning process by proposing a batch size $b$ and do Bayesian update with an averaged

---

[2]We present the results when $|\Theta| = 20, Q = 20$, and we sample $m = 100$ trajectories, evaluated across $n = 100$ problems on school data in the figures here

posterior $\widehat{w}^{(t)} = \frac{\sum_{i \in [b]} w_i^{(t)}}{b}$. Due to space limit, we present the results on the random distribution as follows. Results on other distributions reveal the same conclusion.

We see that our learning method indeed converges to learning the underlying distribution, as the KL divergence is less than $0.025$ and will continue to converge with more learning rounds or a more aggressive learning rate, e.g., $0.5$, see figure 7c in Appendix C. In addition, a more aggressive learning rate tends to make the learning process not so stable, which can be smoothed by increased batch size, as shown in Figures 4b and 4c. Results on more distributions and more learning rates can be found in Appendix C.

## 7. Limitations

Designing a functional data market for machine learning from scratch requires solving many interconnected challenges: discovering useful data for a buyer's task, pricing the value created by that data, designing a usable buyer-facing interaction model, and allocating revenue fairly and efficiently among data sellers. Since solving all of these challenges in one paper would be overly ambitious, we focus on a practical design point: a concrete market design that is simple to use from the buyer side, interface-compatible with existing AutoML services, and able to leverage the value of external data to outperform these services.

At the same time, this scope leaves several limitations. For example, our experiments rely on simulated buyer behavior rather than observations from actual users. In practice, real buyers may deviate from the valuation models assumed in our simulations: they may be more or less price-sensitive, or may interact with the marketplace interface in unexpected ways. Understanding such behavior requires additional, dedicated user studies or deployment-based evaluations. Another limitation concerns the scale of models our market currently supports. Despite the algorithmic efficiency of our data-model discovery engine, repeatedly retraining large models may still be computationally expensive. Supporting such workloads would likely require additional mechanisms, such as model reuse, incremental training, or restricting discovery to cheaper proxy models before validating the most promising options with larger models. We view these problems as important future directions for data-centric ML marketplace research.

## 8. Conclusion

In this paper, we study novel techniques to address core challenges in designing a data-centric machine learning market. With a novel mechanism to price the instrumental value of data, our market offers higher quality models for ML users compared to existing platforms, generates high revenue to compensate participants, all while being API-compatible

and thus practical to use. Recognizing that designing a fully-functional, end-to-end online ML market still requires solving important challenges, we see this work as a step towards the bigger vision of building ML markets.

## Impact Statement

This paper presents work whose goal is to advance the field of Machine Learning. There are many potential societal consequences of our work, none which we feel must be specifically highlighted here.

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

# A. Augmentation and Model Discovery Details

### A.1. Choosing Candidate Augmentation Throughout Iterations

Our data and model discovery approach uses the following mechanisms to choose the best candidate augmentation in each iteration. This approach has two components. i) *candidate generation and likelihood estimation*; ii) *adaptive querying strategy*. The first component identifies the candidate augmentations and computes a feature vector of their properties. The second component ranks the different candidate augmentations and iteratively chooses the best augmentation. This augmentation is given to our model discovery algorithm (line 4 in Algorithm 3).

**Candidate Generation and likelihood estimation.** This component identifies features that can augment $D_{in}$ using database joins. Each candidate feature is processed to compute its vector of data profiles[3]. These profiles are used to cluster candidate augmentations based on their similarity. Intuitively, augmentations in the same cluster are expected to have a similar model performance. Thus, the discovery approach chooses representatives from each cluster for subsequent stages. The profile-based feature vector for each augmentation is also used to calculate a likelihood score, denoting how likely an augmentation would improve model performance.

**Adaptive Querying strategy.** This component uses the identified clusters and their likelihood scores to choose an augmentation for training an ML model. This component interleaves between two complementary strategies (sequential and group querying), which are shown to have varied advantages. The sequential approach estimates the quality of each candidate augmentation and greedily chooses the best option. The group querying strategy considers augmentation subsets of different sizes to train the ML model. Group selection internally relies on the Thompson sampling method to sample an augmentation for each iteration.

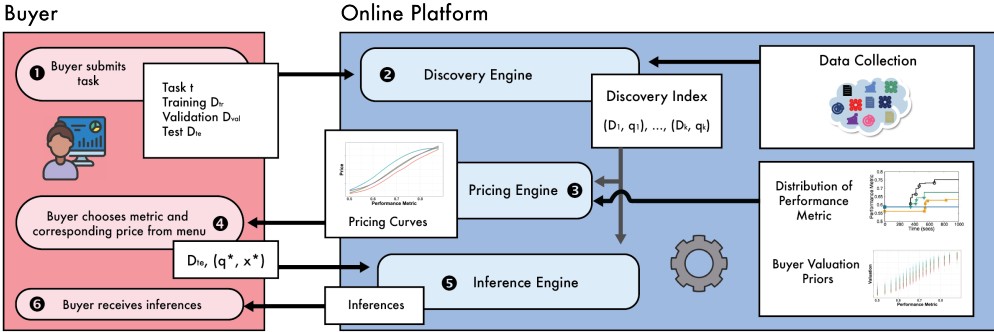

*Figure 5.* Architecture of our market implementation. The discovery engine performs augmentation-model discovery to provide the model qualitys to the pricing engine.

### A.2. Performance Results

To prove the discovery algorithm's approximation ratio, we assume that any augmentation $T$ that improves $M^* \leftarrow D \bowtie T$'s performance the most remains consistent across different ML models.

Note that each model training considers a different ML model for each augmentation. Due to this, a suboptimal augmentation (say $T_1$) may sometimes achieve higher quality than the optimal augmentation $T_2$ because $T_2$ was evaluated with a low-quality ML model. To address this challenge, we consider the top-$t$ tested augmentations again and evaluate their quality with respect to the same model (best model based on the score $u$). In our experiments, choosing $t = \log n$ sufficed to identify the best augmentation, and increasing $t$ helps whenever the set $\mathbf{M}$ has very diverse models with very different quality. Note that our complexity analysis is based on the highest value of $t = n$. We prove the following result about the effectiveness of our approach.

**Proposition A.1.** *Given an initial dataset $D_{in}$, set of augmentations $\mathbf{P}$ and ML models $\mathbf{M}$, our algorithm identifies a*

---

[3]Data profiles refer to properties of these features, e.g., fraction of missing values, correlation with the target column.

*solution $T \subseteq \mathbf{P}$ and $\widehat{M} \in \mathbf{M}$ in $O(\log(|\mathbf{P}|) + |\mathbf{M}|)$ ML model trainings such that*

$$m(\widehat{M} \leftarrow D_{in} \bowtie T) \geq m(M^* \leftarrow D_{in} \bowtie T^*) \times \left( \frac{1}{\alpha}(1 - e^{-\alpha\eta}) - k\epsilon \right)$$

*in cases where the best augmentation for the most powerful model is no worse than other augmentations for other models. $T^*, M^*$ denote the optimal solution and $k, \epsilon$ are constants, $\alpha$ and $\eta$ are curvature and submodularity ratio of $m$.*

*Proof.* Using the result from (Galhotra et al., 2023), the optimal solution after clustering the candidate augmentations is $(1 - k\epsilon)$ an approximation of the overall optimal solution. Further, the sequential querying strategy chooses the best augmentation in each iteration (this holds for any model because the best augmentation according to $M^*$ is also the best according to any other model $M$). Combining these results, we get the following (Galhotra et al., 2023).

$$m(M^* \leftarrow D_{in} \bowtie T) \geq m(M^* \leftarrow D_{in} \bowtie T^*) \times \left( \frac{1}{\alpha}(1 - e^{-\alpha\eta}) - k\epsilon \right) \tag{4}$$

As a last step, Algorithm 3 chooses the best ML model for the returned augmentation $T$, implying $m(\widehat{M} \leftarrow D_{in} \bowtie T) \geq m(M^* \leftarrow D_{in} \bowtie T)$. Combining this with Equation 4 results, we get the desired result. $\qquad \square$

### A.3. Proof of Theorem 4.2

With a common prior distribution $\mu$ over the buyer population, we propose to estimate the Markov chain process $\{\mathbf{P}_t\}_{t \in [T]}$ by sampling quality trajectories from the empirical data discovery process. Specifically, we sample a set of quality trajectories $S$, where $s \in S$ represents a sequence of model qualities discovered in a prefixed period (i.e., $s = [q_1, \cdots, q_T]$). This approach is often more practical in real-world settings where accurate information may be limited or costly to obtain. By utilizing sampled quality trajectories, the approach can provide useful insights and make decisions based on available information, enabling practical and efficient implementations. In addition, by sampling from quality trajectories, the method is inherently robust to uncertainties and noise present in the data. As we shall see in the evaluation section, this robustness allows our algorithm to handle noisy or incomplete data more effectively.

Given a set of quality trajectories $S$, then the market's optimal pricing problem can be computed by the following optimization.

$$\max \sum_{\theta} \mu(\theta) \frac{1}{m} \sum_{s_i \in S} x\big(q^*(\theta, x, s_i)\big) \tag{5}$$

where $q^*(\theta, x, s_i) = \arg\max_{q \in s_i} v^\theta(q) - x(q)$ denotes the buyer type $\theta$'s optimal choice of the model quality under the contract $x$ and sampled quality trajectory $s_i$. As a result, solving the optimal pricing curve involves solving the above complicated bi-level optimization program. Next, we show that this optimization problem can be formulated as Mixed Integer Linear Programming (MILP) which can be solved efficiently by industry-standard solvers such as Gurobi (Gurobi Optimization, LLC, 2022) and Cplex (Cplex, 2009).

We divided the proof of our Theorem 4.2 into two parts. Lemma A.2 showed that the MILP (6) computes the optimal pricing curve $\widehat{x}$ on the sample $\widehat{S} \subseteq S$. Then lemma A.4 showed that using the estimated pricing curve $\widehat{x}$ is an additively $\epsilon$-approximation to the optimal curve $x^*$ on the population $S$. Hence, combining our two lemmas gives us our theorem.

**Lemma A.2.** *Given sampled quality trajectories set $S$, the optimal price curve that maximizes the expected payment (5) can be computed by a MILP.*

$$
\begin{aligned}
\max \quad & \sum_{\theta} \mu(\theta) \sum_{s \in S} \frac{1}{m} \sum_{q \in s} y(\theta, s, q) \\
s.t. \quad & 0 \leq a_{\theta,s} - \big[ v^\theta(q) - x(q) \big] \leq M\big(1 - z(\theta, s, q)\big), \ \forall \theta, s, q \\
& \sum_{q \in s} z(\theta, s, q) = 1, \ \forall \theta, s, \\
& y(\theta, s, q) \leq x(q); \quad y(\theta, s, q) \leq M z(\theta, s, q), \ \forall \theta, s, q \\
& y(\theta, s, q) \geq x(q) - \big(1 - z(\theta, s, q)\big)M, \ \forall \theta, s, q \\
& \boldsymbol{x} \geq 0; \quad \boldsymbol{z} \in \{0, 1\}; \quad \boldsymbol{y} \geq 0.
\end{aligned}
\tag{6}
$$

*Proof.* First of all, we represent the buyer's choice as a binary decision variable $z(\theta, s, q) \in \{0, 1\}$ where $s \in S$ and $q \in s$. $z(\theta, s, q) = 1$ means $q$ is the buyer's optimal choice among all model qualitys in $s$. To model this choice, we propose the following constraint

$$0 \leq a_{\theta,s} - \left[v^\theta(q) - x(q)\right] \leq M\left(1 - z(\theta, s, q)\right) \tag{7}$$

where $a_{\theta,s}$ is a decision variable. Note that under constraint (7), buyer $\theta$'s choice of $q$ where $z(\theta, s, q) = 0$ satisfies $v^\theta(q) - x(q) \leq a_{\theta,s}$ while $z(\theta, s, q) = 1$ satisfies $v^\theta(q) - x(q) = a_{\theta,s}$. As a result, (7) correctly models the buyer's optimal choice and we can rewrite the optimization program as follows

$$
\begin{aligned}
\max \quad & \sum_\theta \mu(\theta) \sum_{s \in S} \frac{1}{m} \sum_{q \in s} x(q) z(\theta, s, q) \\
\text{s.t.} \quad & 0 \leq a_{\theta,s} - \left[v^\theta(q) - x(q)\right] \leq M\left(1 - z(\theta, s, q)\right), \ \forall \theta, s, q; \\
& \sum_q z(\theta, s, q) = 1, \ \forall \theta, s; \\
& \boldsymbol{x} \geq 0; \quad \boldsymbol{z} \in \{0, 1\}.
\end{aligned} \tag{8}
$$

where $x(q) z(\theta, s, q)$ contributes to the expected payment only when $z(\theta, s, q) = 1$. Thus, we have

$$\sum_\theta \mu(\theta) \sum_{s \in S} \frac{1}{m} \sum_{q \in s} x(q) z(\theta, s, q) = \sum_\theta \mu(\theta) \frac{1}{m} \sum_{s_i \in S} x\left(q^*(\theta, x, s_i)\right).$$

However, the above program involves the multiplication of two decision variables, which still cannot be efficiently solved by the industry-standard optimization solvers. Our final step is to linearize the multiplication of these two decision variables by introducing one more variable

$$y(\theta, s, q) = x(q) z(\theta, s, q). \tag{9}$$

In order to linearized the multiplication, we need $y(\theta, s, q) = x(q)$ when $z(\theta, s, q) = 1$, and $y(\theta, s, q) = 0$ otherwise. As a result, we propose the following constraints for linearizing $x(q) z(\theta, s, q)$.

$$
\begin{aligned}
& y(\theta, s, q) \leq x(q); \quad y(\theta, s, q) \leq M z(\theta, s, q), \ \forall \theta, s, q \\
& y(\theta, s, q) \geq x(q) - \left(1 - z(\theta, s, q)\right) M, \ \forall \theta, s, q \\
& \boldsymbol{y} \geq 0
\end{aligned} \tag{10}
$$

where $M$ is a super large constant. Combining (8) – (10) finishes the proof of the proposition, and the MILP (6) has $\boldsymbol{x}$ (of size $|Q|$), $\boldsymbol{y}$ (of size $|\Theta||S||Q|$), $\boldsymbol{z}$ (of size $|\Theta||S||Q|$), and $\boldsymbol{a}$ (of size $|\Theta||S|$) as decision variables. $\qquad \square$

Lemma A.2 shows that given a set of quality trajectories, the price curve that maximizes the expected payment can be computed by MILP. Given a set of finite trajectories $H$ and corresponding valuations $V$, we denote the theoretically optimal profit the market can achieve as $\overline{g}(\boldsymbol{x}^*, S)$ [4], where $S = (H, V)$ and the profit is averaged over all tasks $s \in S$, by solving the program (6). For some buyer and corresponding trajectory $s_i \in S$, we will let $g_i(\boldsymbol{x}, s_i)$ denote the profit the market earns from this buyer when using the pricing scheme. We will assume that the buyer's valuations $v_i$ are bounded by some constant $b$, where $0 \leq v_i \leq b$. Hence, the profit is also bounded, where $0 \leq g_i(\boldsymbol{x}, s_i) \leq b$. Next, we show that by sampling a subset of valuations and trajectories from the set $S$, we can approximate the optimal pricing scheme $\boldsymbol{x}^*$ with high probability when the number of samples is large enough. We first introduce a useful lemma for bounding the deviation of a random variable from its expected value:

**Lemma A.3** (Hoeffding's inequality). *Let $X_1, \cdots, X_m$ be $m$ identical independently distributed samples of a random variable $X$ distributed by $S$, and $a \leq x_i \leq b$ for every $x_i$, then for a small positive value $\epsilon$:*

$$\mathbb{P}\left[\mathbb{E}[X] - \tfrac{1}{m} \sum_i X_i \geq \epsilon\right] \leq \exp\left(\tfrac{-2m\epsilon^2}{(b-a)^2}\right)$$

*and*

$$\mathbb{P}\left[\mathbb{E}[X] - \tfrac{1}{m} \sum_i X_i \leq -\epsilon\right] \leq \exp\left(\tfrac{-2m\epsilon^2}{(b-a)^2}\right)$$

---

[4]The objective value also depends on $\boldsymbol{y}$ and $\boldsymbol{z}$, which is implicitly decided through the constraints of (6) once $\boldsymbol{x}$ is given. For the sake of simplicity in notation, we omit to mention these two variables since $x$ is the only decision variable for the market

Next, we present the last step of the proof our theorem 4.2:

**Lemma A.4.** *Let $S$ be the total set of possible tasks (trajectories and valuations) that the market might encounter. With probability $1 - 2\delta$ over the draw of samples $\widehat{S}$ of $m = |\widehat{S}|$ samples from $S$, we can compute the pricing scheme $\widehat{x}$ such that*

$$\overline{g}(\widehat{x}, S) \geq \overline{g}(x^*, S) - \epsilon.$$

*where $\delta = \exp(\frac{-2m\epsilon^2}{b^2})$, $b$ is the maximum valuation from any buyer, and $\epsilon > 0$ is the error term.*

*Proof.* Given any independently sampled set $\widehat{S}$ from the total set $S$ of trajectories, we let $\widehat{x}$ denote the *optimal* solution to (6) when we are maximizing $\overline{g}(x, \widehat{S})$ over the sample set. Hence, both $g(\widehat{x}, s)$ and $g(x^*, s)$ are random variables for some sample $s \in S$, and $\overline{g}(\widehat{x}, \widehat{S})$ and $\overline{g}(x^*, \widehat{S})$ are sample means. Moreover, since we are random sampling, the expected value of $g(x, s)$ is just the population mean, where $\mathbb{E}[g(x, s)] = \overline{g}(x, S)$.

By instantiating Lemma A.3, we have that with probability $1 - \exp\left(\frac{-2m\epsilon^2}{b^2}\right)$,

$$\overline{g}(\widehat{x}, S) = \mathbb{E}[g(x, s)] \geq \frac{1}{m} \sum_s g(\widehat{x}, s) = \overline{g}(\widehat{x}, \widehat{S}) - \epsilon \tag{11}$$

Moreover, by definition that $\widehat{x}$ is the optimal solution to (6) over the sample set $\widehat{S}$ (ie. it maximizes $\overline{g}(x, \widehat{S})$), we get $\overline{g}(\widehat{x}, \widehat{S}) \geq \overline{g}(x^*, \widehat{S})$. Furthermore, by instantiating Lemma A.3 again, we know that with probability $1 - \exp\left(\frac{-2m\epsilon^2}{b^2}\right)$,

$$\overline{g}(x^*, \widehat{S}) = \frac{1}{m} \sum_s g(x^*, s) \geq \mathbb{E}[g(x^*, s)] - \epsilon = \overline{g}(x^*, S) - \epsilon \tag{12}$$

Then, using an union bound on the probability of failure $\delta = \exp(\frac{-2m\epsilon^2}{b^2})$ for equations (11) and (12), we know the probability of both of them not holding is at max $2\delta$.

Hence, combining equations (11) - (12), we have

$$\overline{g}(\widehat{x}, S) > \overline{g}(\widehat{x}, \widehat{S}) - \epsilon \geq \overline{g}(x^*, \widehat{S}) - \epsilon > \overline{g}(x^*, S) - 2\epsilon$$

with probability $1 - 2\delta = 1 - 2\exp(\frac{-2m\epsilon^2}{b^2})$. $\qquad\square$

**Remark A.5.** *Given any $\delta$, we can achieve any approximation error $\epsilon$ by sampling $m = -\frac{1}{2}\log(\delta)\frac{b^2}{\epsilon^2}$ samples from $S$ to form $\widehat{S}$.*

# B. Omitted Proofs from Section 5

The proof of Theorem 5.2 is divided into two parts.

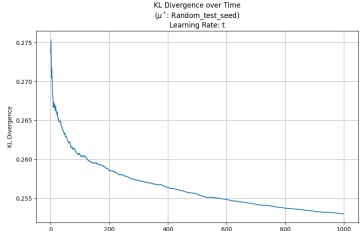
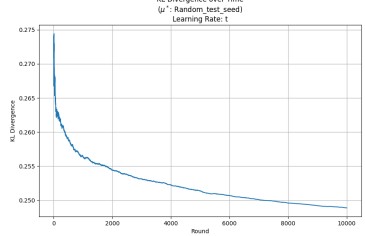
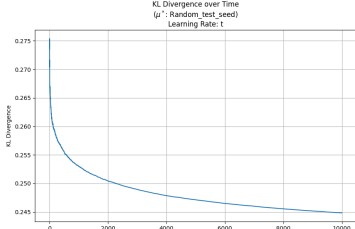

*(a)* Batchsize is 100, number of rounds is 1000.

*(b)* Batchsize is 10, number of rounds is 10000.

*(c)* Batchsize is 100, number of rounds is 10000.

*Figure 6.* The underlying prior distribution $\mu^* \in \Delta^n$ is a randomly generated distribution, learning rate $\eta = 1/(t+1)$.

**Bounding the revenue loss under approximately optimal prior** With $||\mu - \mu^*||_2 \leq \epsilon$, we get $\sum_\theta \left|\mu(\theta) - \mu^*(\theta)\right| \leq \sqrt{n}||\mu - \mu^*||_2 = \sqrt{n}\epsilon$. We show that given an optimal solution to (6) with respect to true $\mu^*$, it approximates the optimal solution under $\mu$ as well. Suppose the price curve $x(q) \in [\underline{p}, \overline{p}], \forall q$, then we can bound the objective difference by

$$\left|\sum_\theta \mu(\theta)x(\mu, \theta) - \mu^*(\theta)x(\mu^*, \theta)\right| \leq \overline{p}\sum_\theta \left|\mu(\theta) - \mu^*(\theta)\right| \leq \overline{p}\sqrt{n}\epsilon \tag{13}$$

**Bounding the revenue loss under approximate transition matrix $P$.** We define $\tau^*(P)$ as the buyer's optimal stopping policy computed by Algorithm 1 with Markov transition matrix $P$. In addition, we define $\Phi^{\tau^*(P)}(\cdot, \cdot, \cdot)$ as the buyer's expected utility, i.e., the corresponding 3-dimensional DP table from Algorithm 1. In addition, let $\overline{V} = \max_{\theta,q} v^\theta(q)$, which is the maximal achievable utility for any agent type and model quality. We have $\Phi^{\tau^*(P)}(q_1, q_2, t) \leq \overline{V}$ for any $q_1, q_2, t, P$. First of all, we compare $\Phi^{\tau^*(P)}(0, 0, 0)$ and $\Phi^{\tau^*(\widehat{P})}(0, 0, 0)$.

- When $t = T$, we have $\Phi^{\tau^*(P)}(q_1, q_2, T) = \Phi^{\tau^*(\widehat{P})}(q_1, q_2, T)$

- When $t = T - 1$, we have

$$\Phi^{\tau^*(P)}(q_1, q_2, t) = \max\left(v^\theta(q_1) - x(q_1) - tc,\right.$$
$$\left.\mathbb{E}_{P(q|q_2)}\left[\Phi(\operatorname*{argmax}_{\{q_1,q\}}(v^\theta(q_1) - x(q_1) - c \cdot t, v^\theta(q) - x(q) - (t+1) \cdot c), q, t+1)]\right)\right.$$

$$\Phi^{\tau^*(\widehat{P})}(q_1, q_2, t) = \max\left(v^\theta(q_1) - x(q_1) - tc,\right.$$
$$\left.\mathbb{E}_{\widehat{P}(q|q_2)}\left[\Phi(\operatorname*{argmax}_{\{q_1,q\}}(v^\theta(q_1) - x(q_1) - c \cdot t, v^\theta(q) - x(q) - (t+1) \cdot c), q, t+1)]\right)\right.$$

To begin with, we provide a bound for

$$\left|\mathbb{E}_{P(q|q_2)}\left[\Phi^{\tau^*(P)}(\operatorname*{argmax}_{\{q_1,q\}}(v^\theta(q_1) - x(q_1) - c \cdot t, v^\theta(q) - x(q) - (t+1) \cdot c), q, T)\right]\right.$$
$$\left.- \mathbb{E}_{\widehat{P}(q|q_2)}\left[\Phi^{\tau^*(\widehat{P})}(\operatorname*{argmax}_{\{q_1,q\}}(v^\theta(q_1) - x(q_1) - c \cdot t, v^\theta(q) - x(q) - (t+1) \cdot c), q, T)\right]\right| \leq$$
$$\sum_q |P(q|q_2) - \widehat{P}(q|q_2)| \cdot \Phi^{\tau^*(P)}(\operatorname*{argmax}_{\{q_1,q\}}(v^\theta(q_1) - x(q_1) - c \cdot t, v^\theta(q) - x(q) - (t+1) \cdot c),$$
$$q, T) \leq \epsilon\sqrt{k\overline{V}}$$

Next, note that $\max(\cdot, \cdot)$ is 1-Lipschitz, we have

$$|\Phi^{\tau^*(P)}(q_1, q_2, t) - \Phi^{\tau^*(\widehat{P})}(q_1, q_2, t)| \leq$$
$$\left|\mathbb{E}_{P(q|q_2)}\left[\Phi^{\tau^*(P)}(\operatorname*{argmax}_{\{q_1,q\}}(v^\theta(q_1) - x(q_1) - c \cdot t, v^\theta(q) - x(q) - (t+1) \cdot c), q, T)\right] -\right.$$
$$\left.\mathbb{E}_{\widehat{P}(q|q_2)}\left[\Phi^{\tau^*(\widehat{P})}(\operatorname*{argmax}_{\{q_1,q\}}(v^\theta(q_1) - x(q_1) - c \cdot t, v^\theta(q) - x(q) - (t+1) \cdot c), q, T)\right]\right|$$
$$\leq \epsilon\sqrt{k\overline{V}}$$

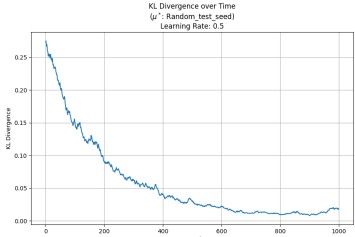

*(a)* Batchsize is 100, number of rounds is 1000.

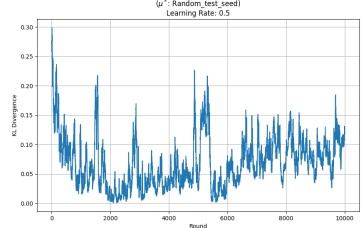

*(b)* Batchsize is 10, number of rounds is 10000.

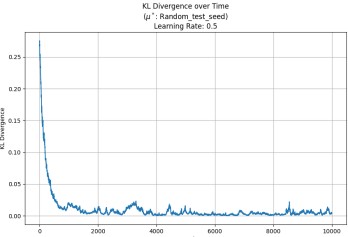

*(c)* Batchsize is 100, number of rounds is 10000.

*Figure 7.* The underlying prior distribution $\mu^* \in \Delta^n$ is a randomly generated distribution, learning rate $\eta = \frac{1}{2}$.

- When $t = T - 2$, we have

$$\Phi^{\tau^*(\boldsymbol{P})}(q_1, q_2, t) = \max \Big( v^\theta(q_1) - x(q_1) - tc,$$

$$\mathbb{E}_{\boldsymbol{P}(q|q_2)} \big[ \Phi^{\tau^*(\boldsymbol{P})}(\underset{\{q_1, q\}}{\operatorname{argmax}}(v^\theta(q_1) - x(q_1) - c \cdot t, v^\theta(q) - x(q) - (t+1) \cdot c), q, T-1)] \Big)$$

$$\Phi^{\tau^*(\widehat{\boldsymbol{P}})}(q_1, q_2, t) = \max \Big( v^\theta(q_1) - x(q_1) - tc,$$

$$\mathbb{E}_{\widehat{\boldsymbol{P}}(q|q_2)} \big[ \Phi^{\tau^*(\widehat{\boldsymbol{P}})}(\underset{\{q_1, q\}}{\operatorname{argmax}}(v^\theta(q_1) - x(q_1) - c \cdot t, v^\theta(q) - x(q) - (t+1) \cdot c), q, T-1)] \Big)$$

Similarly, we provide a bound for

$$\Big| \mathbb{E}_{\boldsymbol{P}(q|q_2)} \big[ \Phi^{\tau^*(\boldsymbol{P})}(\underset{\{q_1, q\}}{\operatorname{argmax}}(v^\theta(q_1) - x(q_1) - c \cdot t, v^\theta(q) - x(q) - (t+1) \cdot c), q, T-1)] -$$

$$\mathbb{E}_{\widehat{\boldsymbol{P}}(q|q_2)} \big[ \Phi^{\tau^*(\widehat{\boldsymbol{P}})}(\underset{\{q_1, q\}}{\operatorname{argmax}}(v^\theta(q_1) - x(q_1) - c \cdot t, v^\theta(q) - x(q) - (t+1) \cdot c), q, T-1)] \Big| =$$

$$\Big| \sum_q \boldsymbol{P}(q|q_2) \cdot \Phi^{\tau^*(\boldsymbol{P})}(\underset{\{q_1, q\}}{\operatorname{argmax}}(v^\theta(q_1) - x(q_1) - c \cdot t, v^\theta(q) - x(q) - (t+1) \cdot c), q, T) -$$

$$\widehat{\boldsymbol{P}}(q|q_2) \cdot \Phi^{\tau^*(\widehat{\boldsymbol{P}})}(\underset{\{q_1, q\}}{\operatorname{argmax}}(v^\theta(q_1) - x(q_1) - c \cdot t, v^\theta(q) - x(q) - (t+1) \cdot c), q, T) \Big|$$

$$\leq 2\epsilon\sqrt{kV}$$

By induction, we have $|\Phi^{\tau^*(\boldsymbol{P})}(0,0,0) - \Phi^{\tau^*(\widehat{\boldsymbol{P}})}(0,0,0)| \leq T\epsilon\sqrt{kV}$.

Next, we provide a bound for the performance of a specific policy $\pi^*(\widehat{\boldsymbol{P}})$ under different Markov transition matrices $\widehat{\boldsymbol{P}}$ and $\boldsymbol{P}$. We denote $f\big(\pi^*(\widehat{\boldsymbol{P}}), \boldsymbol{P}, 0, 0, \big)$ as the buyer's expected utility under $\boldsymbol{P}$ while they are adapting the policy $\pi^*(\widehat{\boldsymbol{P}})$ computed from $\widehat{\boldsymbol{P}}$. Note that we have $\Phi^{\tau^*(\boldsymbol{P})}(0,0,0) = f\big(\pi^*(\boldsymbol{P}), \boldsymbol{P}, 0, 0, 0\big)$.

At each state $(q_1, q_2, t)$, if the policy decides to stop, then the policy has the same expected utility under two different Markov transition matrices, i.e., $f\big(\pi^*(\widehat{\boldsymbol{P}}), \boldsymbol{P}, q_1, q_2, t\big) = f\big(\pi^*(\widehat{\boldsymbol{P}}), \widehat{\boldsymbol{P}}, q_1, q_2, t\big)$; Otherwise, we have $|f\big(\pi^*(\widehat{\boldsymbol{P}}), \boldsymbol{P}, q_1, q_2, t\big) - f\big(\pi^*(\widehat{\boldsymbol{P}}), \widehat{\boldsymbol{P}}, q_1, q_2, t\big)| \leq |\Phi^{\tau^*(\widehat{\boldsymbol{P}})}(q_1, q_2, t) - \Phi^{\tau^*(\boldsymbol{P})}(q_1, q_2, t)|$. Therefore, we have $|f\big(\pi^*(\widehat{\boldsymbol{P}}), \boldsymbol{P}, 0, 0, 0\big) - f\big(\pi^*(\widehat{\boldsymbol{P}}), \widehat{\boldsymbol{P}}, 0, 0, 0\big)| \leq |\Phi^{\tau^*(\boldsymbol{P})}(0,0,0) - \Phi^{\tau^*(\widehat{\boldsymbol{P}})}(0,0,0)| \leq T\epsilon\sqrt{kV}$ as well.

Finally, combing the above two processes, we have

$$|f\big(\pi^*(\boldsymbol{P}), \boldsymbol{P}, 0, 0, 0\big) - f\big(\pi^*(\widehat{\boldsymbol{P}}), \boldsymbol{P}, 0, 0, \big)| \leq$$

$$|f\big(\pi^*(\boldsymbol{P}), \boldsymbol{P}, 0, 0, 0\big) - f\big(\pi^*(\widehat{\boldsymbol{P}}), \widehat{\boldsymbol{P}}, 0, 0, \big)| + |f\big(\pi^*(\widehat{\boldsymbol{P}}), \widehat{\boldsymbol{P}}, 0, 0, 0\big) - f\big(\pi^*(\widehat{\boldsymbol{P}}), \boldsymbol{P}, 0, 0, \big)| =$$

$$|\Phi^{\tau^*(\boldsymbol{P})}(0,0,0) - \Phi^{\tau^*(\widehat{\boldsymbol{P}})}(0,0,0)| + |f\big(\pi^*(\widehat{\boldsymbol{P}}), \widehat{\boldsymbol{P}}, 0, 0, 0\big) - f\big(\pi^*(\widehat{\boldsymbol{P}}), \boldsymbol{P}, 0, 0, \big)| \leq 2T\sqrt{kV}\epsilon$$

## C. Omitted Results from Section 6

### C.1. Additional Experimental Results From RQ3

Due to space limitations, we show additional experimental results (Figures 6, 7) from RQ3, which apply different learning rates to the same prior distribution. We have run more experiments on wider settings of prior distributions, and have observed similar behaviours across those settings, so we believe the following plots are representative.

