# OpenReview forum: "Optimal Pricing for Data-Augmented AutoML Marketplaces"
_ICML.cc/2026/Conference — ICML 2026 regular_

### Official Review · Reviewer_jud7 · 2026-03-09

**Soundness:** 3
**Presentation:** 2
**Significance:** 2
**Originality:** 2
**Overall Recommendation:** 4
**Confidence:** 3

**Summary:**

Authors propose a data-augmented autoML platform that automatically augments buyer-submitted training data with valuable external datasets, pricing the resulting models based on their measurable performance improvements (instrumental value), and sends inferences back to the buyer.

**Compliance With Llm Reviewing Policy:**

Affirmed.

**Final Justification:**

Authors responded to questions I and other reviewers had, and promised to make the relevant changes, so I raised my score.

**Key Questions For Authors:**

Most of my questions are specific to the theoretical modeling and the alignment between the motivation and the proposed platform. Please see the weakness section for all the concerns I have and an elaboration of some of the questions I list below.

- How does the platform handle misreporting and negative utility?

- What kind of data do buyers send to the online platform? Please see weakness 2 for more information on this question.

- Why is it that sometimes the realized surplus at time t depends on t and sometimes it doesn’t?

**Limitations:**

No. Although there are lots of limitations of the proposed platform, authors didn't include a limitations/discussion section in the main paper of appendix.

**Strengths And Weaknesses:**

*Strengths*

The authors propose a data-augmented AutoML marketplace that supports instrumental value–based pricing of datasets and helps buyers identify both suitable data and ML models for their tasks. They evaluate the platform using the NYC Open Data, which contains 69K datasets.

Although the authors initially assume knowledge of the prior distribution over buyers’ private valuations, they also examine how a seller can learn this prior over time through repeated interactions with buyers. In addition, they quantify the effect of using an approximately optimal prior by deriving bounds on the resulting revenue loss.



*Weaknesses*

* There is conflict between lines 22-34 (RHS) ``... augments buyer’s initial training data with additional features on the platform ...`` and lines 90-93 (RHS) ``....The buyer may also come with some initial training data, which does not affect our design....`` This is confusing to me. Appendix Figure 4 also appears to indicate that the buyer submits the task along with three datasets: training, validation, and testing. Could the authors clarify whether this interpretation is correct?

- In line 92 (RHS), the authors define q as the performance metric, and in 104-108, they indicate that q changes with time. How diverse/different are these metrics?  To be certain, is the model metric synonymous with the performance metric? What does rounded metrics mean?

* The buyer’s quasi-linear utility depends on the buyer’s valuation of a given performance metric, the price charged for that metric, and the search cost incurred per unit of time. Because the buyer’s valuation of the performance metric is private and individually determined, it is unclear to me how the platform addresses negative utility that could arise from the buyer (fraudulently) valuing the metric(s) so low and or much lower than the price charged for that metric.

- In line 180, the authors define \phi(q, q_t, t) as the optimal future reward the buyer can achieve in state
(q, q_t, t) under an optimal policy. However, in line 192, the realized q is defined independently of t. Could the authors clarify this?

* In general, the theory sections require lots of effort to follow due to a combination of missing or unclear definitions, overloaded notation, and presentation issues. For instance, line 175 may be better formatted as an equation instead of as an inline expression. In addition, using q to denote the maximal surplus at time t and q_t to denote the realized surplus at time t can be confusing. It would also be helpful to explicitly state the domain or range of the key variables.

- The authors argue that buyers are unlikely to manipulate because doing so ``requires a high degree of expertise from the buyer, which is atypical of the targeted users of our AutoML market.`` However, computing \tau^\star appears to be nontrivial. If the framework assumes that buyers are capable of determining \tau^\star, then it seems plausible that they may also possess the incentive and knowledge needed to engage in strategic manipulation.

* MILP is a computationally intensive approach. Combined with the inherent cost of model training and evaluation, I am uncertain about the overall scalability of the proposed platform.

- If buyer-contributed data is unintentionally adversarial, inherently more difficult to classify, or drawn from a minority population that is often underrepresented in model training, overall model performance/quality may decline. In many cases, this decline can reflect valuable information about model deficiencies. However, this framework would penalize the contributors for this and undervalue the data, similar to the status quo.

* I am uncertain whether the proposed framework fully addresses the stated motivations or offers practical advantages over existing AutoML platforms. In particular, it is unclear whether the data-augmentation component would meaningfully increase participation or improve transparency in how data is priced. For instance, the framework may not resolve the challenge highlighted by the authors: ``Many organizations that would benefit from machine learning lack access to the necessary training data, which is costly to obtain.`` In the proposed setting, buyers submit their own training data to the platform and pay for the resulting model based on the estimated improvement in quality. This implicitly assumes that buyers already possess sufficiently relevant or high-quality data for the task.

- ``(i) they typically price data based on the raw dataset itself, rather than on the actual utility it provides through a trained machine learning model; (ii) they often lack transparency in how prices are determined.`` While I do agree that this could be an issue, pricing data based on utility undercuts the intrinsic value of the data. A more effective approach would price data based on both its intrinsic value and its extrinsic (instrumental) value (Mitchell et al., 2022, *Measuring Data*; Castro Fernández, 2025, *What is the Value of Data?: A Theory and Systematization*). Even under such a framework, clear documentation and transparency specific to the data valuation and pricing process would be essential so that all stakeholders understand how prices are determined. In my opinion, simply providing visibility into how model performance evolves as buyers decide when to stop the data and model search is not by itself a sufficient form of transparency for price determination.





*Miscellaneous*

* There might be some parallels to work on the Pandora’s box gittins index.
- Some equations in the margin, such as the one on line 230, are cramped. Try using \\! to reduce spacing, and if possible, define variables before they appear in the equation.
- Algorithm 2 is so compressed that there is no space between it and the text below and above.
* Improve the font size of the results figures. For example, text in Appendix Figures 5 and 6 is barely readable.

---

> ### Author Rebuttal · Authors · 2026-03-31
>
> We thank the reviewer for the great feedback. We answer the major concerns below, and hope it will show the value of our work more clearly.
>
> 1. What does buyer send
>
> Correct, buyers send in training, validation, and testing data to the platform. We will clarify this in the paper.
>
> 2. Misreporting/Negative Utility
>
> Excellent question. We start by noting that these cases are rare: for our market to have no improvement, it must be that no single dataset in the entire repo aligns well with the buyer task. In RQ1, we have shown that across 1000 different tasks, our repo brings meaningful improvements. In rare cases when there's no improvement, our market adopts the same flow: it presents a menu with the discovered model-quality pairs. For buyers, they observe that there are no augmentations leading to improvement, and may realize that their data is unintentionally adversarial. They may choose to leave the market, or decided to examine those augmentations with negative utility further - totally up to them. An important difference between our market and existing ones are that, in existing markets, there is no way for buyers to even determine the value of a dataset with respect to a task they care about. In ours, even in negative utility cases, buyers still get information (e.g. their validation set is niche). They can then leverage this information to their advantage and take according actions. In this sense, we are not really "penalizing the seller", but revealing how well their datasets align with a particular task buyer cares about.
>
> 3. Clarifying "q" and "t"
>
> Thank you for pointing this out. q_t denotes the currently realized performance metric at time t, and q denotes the best-so-far metric observed up to time t, so q is implicitly time-dependent. We will fix this confusion.
>
> 4. Meaning of the "Metrics"
>
> Yes, all these metrics refer to the performance of the model, with respect to a buyer specified notion of "metric". One example is the classification accuracy on a validation set, which is a number. We will standardize these.
>
> 5. Solution Scalability and System Cost
>
> We kindly refer to our answer to Reviewer o5tn. In summary, over a repository of 70K datasets with 30M columns and 3B rows, our solution reaches convergence between 30 and 50 minutes, on average for 1000 tasks. The complexity for MILP is in the order of O(|# performance metrics| * |# buyer types| * |# trajectories|). And our experiment settings are solved efficiently with existing solvers. These evidence show the scalability of our design.
>
> 6. Does our pricing scheme and data-augmentation increase participation?
>
> If we take away all datasets from our market, it degenerates into the same architecture as existing AutoML platforms. By leveraging additional datasets, we strictly outperform existing solutions, as long as there exists some relevant datasets to the buyers' needs, and even a few additional features can achieve greater performance. In RQ1, we have shown that our market beats plain AutoML solutions significantly over many different tasks. Furthermore, even in existing AutoML platforms, buyers need to bring in their training dataset, so this is not an additional bottleneck introduced by our design - our market is designed to be  interface-compatible to existing solutions.
>
> We believe this design brings benefits to both sides of the market: for buyers, they face lower risk because as long as some datasets on the market are relevant to their task, they will extract value from the data; for sellers, learning from buyer trajectories for pricing also gives them a much safer way to profit from their data without losing much value, because buyers are more willing to make a purchase knowing the seller data is useful. Comparing this to existing data markets where sellers just "fix a price": when priced too high, buyers have little motivation to buy, since they have little insights into its usefulness. When priced too low, sellers may be severely exploited. This is actually the main bottleneck in existing data markets like AWS or Snowflake. Reusing examples above: it would be hard for buyer and seller to agree on a monthly IMDB subscription for 150,000 USD, by just showing them the dataset schema.
>
> 7. Strategic manipulation
>
> Excellent question. The best way to think about this is that we target non-experts - the same audience as existing AutoML users. For these users, they can benefit from our pricing schema even without solving the DP exactly: they can observe the trajectory evolution for a little bit, and pick one from the menu that best fits their needs. Solving it exactly only helps them further. It's true that highly-skilled buyer can still manipulate the market. Yet, entire prevention of strategic actions is extremely difficult, which we believe is out of scope of this paper.
>
> Finally, we thank the reviewer for pointing out other inconsistencies. We will incorporate those changes, and hope the above analysis better shows our contributions.

---

> > ### Author Rebuttal · Reviewer_jud7 · 2026-04-01
> >
> > Authors have responded to the questions I had. However, most of the responses (mine and those raised by other reviewers, e.g., confusion of the use of the phrase metrics) require heavily editing the paper. I will raise my score in the hope that authors make those edits and include a comprehensive limitations section before camera ready.

---

> > > ### Author Response · Authors · 2026-04-01
> > >
> > > We appreciate the reviewer for taking the time and effort to acknowledge our rebuttal, and will do our due diligence to edit the paper to clarify the confusion, as well as adding a limitation section by moving some other technical details to the Appendix. We are confident that we will be able to make space for the required edits in the updated version of the paper, since those do not need much additional work, and do not concern the core of our technical framework and foundation.

---

### Official Review · Reviewer_o5tn · 2026-03-11

**Soundness:** 2
**Presentation:** 3
**Significance:** 2
**Originality:** 3
**Overall Recommendation:** 4
**Confidence:** 4

**Summary:**

This paper studies pricing in a data-augmented AutoML marketplace, where the platform charges buyers based on model performance improvement induced by external data augmentation. The paper presents an integrated framework including buyer-side optimal stopping, menu-based pricing, prior learning over buyer types, and an augmentation discovery module.  Experiments on a large NYC open-data market suggest improvements in discovery and simulated revenue.

**Compliance With Llm Reviewing Policy:**

Affirmed.

**Final Justification:**

The rebuttal provides additional clarification on the main points raised in my review. While some aspects could still benefit from further refinement, the response helps clarify the authors’ position and addresses the key issues to a reasonable extent. I have updated the corresponding assessment accordingly.

**Key Questions For Authors:**

1. Can the authors provide a clearer end-to-end complexity and system-cost analysis of the full pipeline?
2. Could the authors discuss the main limitations of the current framework, together with possible directions for future work?

**Limitations:**

The authors do not discuss the limitations in the manuscript.

**Strengths And Weaknesses:**

**Strength**
1. The paper addresses an interesting problem at the intersection of data markets, AutoML, and pricing.
2. The framework is relatively complete, covering pricing, buyer behavior, prior learning, and discovery in a unified formulation.
3. The paper is well-organized and easy to follow.

**Weakness**
1. The paper mainly combines several existing ingredients, while the main technical novelty is not clearly isolated. The contribution appears stronger at the system-integration level than at the level of new algorithmic or theoretical insight.
2. The  poposed framework relies on assumptions such as Markovian performance evolution and reliable transition estimation from historical trajectories. The realism of these assumptions is not sufficiently justified.
3.  The paper does not provide a clear end-to-end complexity or deployment-cost analysis for the full pipeline.
4. The experimental evaluation should be strengthened. First, the experiments are conducted on only one data market, which limits the evidence for generalizability across domains and marketplace settings. Second, the paper does not compare with sufficiently representative state-of-the-art baselines, weakening the claim that the proposed method clearly outperforms existing alternatives. Third, the empirical study relies largely on simulated buyer behavior, which is insufficient to support the practical effectiveness of the proposed framework.

---

> ### Author Rebuttal · Authors · 2026-03-31
>
> We thank the reviewer for the great feedback. We start with answering the key questions, then address the other concerns.
>
> 1. Detailed Complexity/System-cost Analysis
>
> As illustrated in the Architecture diagram in the Appendix, the system's cost comes from the discovery engine, and the pricing engine. The joint data-model discovery algorithm is the primary online cost, and it converges in O(log(p)) iterations where P is in the number of augmentations. In our experiments, we have shown that over a repository of 70K datasets with 30M columns and 3B rows, it reaches convergence between 30 and 50 minutes, on average for 1000 different regression and classification tasks. These numbers give empirical evidence to the practicality of the discovery engine. On the other hand, the pricing engine is primarily an offline cost, and gets dominated by the discovery engine: the platform logs metric trajectories (minimal), estimates the transition process (minimal), and periodically solves a sampled MILP to update the price curve. Once the price curve is posted, the online pricing path consists mainly of metric-to-price lookup and logging buyer actions, which is negligible. The offline cost are solving the MILP for pricing curve updates, which is in the order of O(|number of performance metrics| * |number of buyer types| * |number of trajectories|). In our experiments, we used 20 metrics, 20 buyer types, and 100 trajectories, and this is done within hours on industrial grade solvers. This cost will not blow up, because the space for metrics and types are stable, and its cost are amortized over the collection over many trajectories.
>
> Together, we believe the above analysis shows the feasibility of our design.
>
> 2. Limitations and Future Directions
>
> This is a great question, and we will make space to add these in the revised version. One main limitation of the paper is precisely pointed out by W3, which is that more experiments can be added, and it would be better to study actual user behavior rather than simulated ones. Yet, we want to note that this limitation of the paper primarily comes from how large the problem of "designing a functional data market from scratch is". There are myriad, interconnected challenges that need to be solved together: from discovery, to pricing, to a good interaction design, to fair and efficient revenue allocation. It would be very ambitious, if possible at all, to address the entire problem in one go. Yet, we would need to start kicking the tires from some point to advance on the broader data market research. Therefore, in this work we picked a most practical design: one that is simple to use from the buyer side, interface-compatible with existing technologies, yet leverage the value of data to outperform existing services, which is the major strength and novelty of this work. And we designed all these pieces to synergize together well, as shown in our existing experiments. We agree that many questions remain unanswered. Just to name a few: how would deviations of actual buyers' behaviors from simulated ones impact our scheme? How does the performance change, between more popular sections where there are more datasets, and those with fewer datasets? How would an alternative interface affects the revenue extracted, if at all? All these are great directions to explore. With the given space of this paper in itself, we do not claim, or try to answer all these questions at once. But these are excellent candidates for future work, and we plan to discuss this in the paper. Some of these may even require different methodologies (such as a dedicated user study in itself to study user behaviors, or alternative interface design).
>
> Finally, we hope the above discussion give more insights into the technical significance of our work. If we forget about the paper entirely, and start blank state: how would we leverage data in a practical market? We clearly realize the massive design space that is available, and there's a huge gap between what could be done, and the existing markets that are weak, have arbitrary pricing, and have high risk. We have built a deployable system that works with AutoML and automatically identifies augmentation among large volumes of data, and works in a feedback loop with the pricing module. Our design reduces opportunities for strategic behavior (few buyer inputs) and reduce risk with the introduction of instrumental value. To leverage data while reducing buyer inputs, we discover the best data-model pair , which is challenging for large repos. Our pricing engine handles unknown model metric evolution with only access to sample data, and gives buyers flexibility to make purchase decisions-a complicated bi-level optimization.
>
> While this work's contribution may appear not as traditional, we hope to illustrate the value of providing a feasible system that solving the above, interconnected challenges that enables further data market research.

---

> > ### Author Rebuttal · Reviewer_o5tn · 2026-04-02
> >
> > Thank you to the authors for the clarifications. I will reconsider my score.

---

> > > ### Author Response · Authors · 2026-04-02
> > >
> > > We thank the reviewer for taking the time and effort to read our reply, and we are glad that our analysis has helped clarify any remaining concerns.

---

### Official Review · Reviewer_YR6p · 2026-03-12

**Soundness:** 3
**Presentation:** 3
**Significance:** 3
**Originality:** 3
**Overall Recommendation:** 5
**Confidence:** 3

**Summary:**

This paper proposes a novel and practical data-augmented AutoML marketplaces design, whose core lies in pricing based on the marginal improvement in model performance attributable to external data (i.e., instrumental value), rather than traditional computational cost-based pricing. This approach aims to address existing challenges in data markets concerning data pricing, transparency, and the integration of data augmentation. The paper provides a solid theoretical foundation and experimental validation of its market design, pricing mechanism, and prior distribution learning.

**Compliance With Llm Reviewing Policy:**

Affirmed.

**Final Justification:**

The authors’ rebuttal has resolved my concerns. Considering both the original paper and the rebuttal, I believe the current rating is appropriate.

**Key Questions For Authors:**

As shown in weaknesses.

**Limitations:**

yes

**Strengths And Weaknesses:**

**Strengths**
* S1. The contributions of this work are significant. It not only offers a theoretically rigorous pricing framework but also emphasizes its practicality, enabling seamless integration with existing cloud-based AutoML platforms. This paradigm shift, where data is valued for its role in improving model performance rather than as raw data, has important implications for both data markets and the broader AutoML field.
* S2. The paper is indeed theoretically grounded. The theories elucidated within provide a robust theoretical underpinning for the proposed pricing strategy and are complemented by appropriate proofs.
* S3. The paper provides a comprehensive experimental validation. The proposed strategy was tested in a market containing 69K datasets , demonstrating its effectiveness. In addition, the study discusses the revenue implications of different pricing approaches.

**Weaknesses**
* W1. The paper assumes that the data discovery cost $c_t$ is negligible relative to the model price $x(q)$ and the buyer’s valuation. While this simplification facilitates theoretical analysis, in practical scenarios, the data discovery cost may not be directly negligible.
* W2. Different initial user priors may affect learning and convergence, and providing further analysis or justification on this aspect would strengthen the paper’s persuasiveness. Furthermore, incorporating comparisons with conventional pricing strategies into the experimental evaluation would help substantiate the effectiveness of the proposed instrumental value pricing method.

---

> ### Author Rebuttal · Authors · 2026-03-30
>
> We thank the reviewer for the great feedback. Here are our thoughts on the W1 and W2.
>
> For W1: Assumption on negligible discovery cost
>
> Indeed, in practical scenarios, discovery cost may not be negligible, and more so for buyers with low valuations of the task. However, the discovery cost is still minimal, in comparison to both the current AutoML platform charges. For example, on the cloud, a g5.xlarge instance with 4 vCPUs, 16 GB of memory is priced at 1 USD per hour for compute. This is little comparing to existing AutoML services: Google's Vertex AI charges 21 USD per hour, for tabular data ML services. The compute cost becomes even smaller for buyers with larger valuations of the task (e.g. large corporates). Therefore, we believe this simplification is reasonable to make in the paper.
>
> For W2: Additional Details on Prior and Pricing
>
> Different user priors have an impact on learning and convergence. In the paper, we have validated our learning method by running over various distributions, including a distribution that’s generated randomly, a uniform distribution, a slightly skewed distribution, a highly skewed distribution, and an extremely skewed distribution. We observed that our learning algorithm indeed converges under these different priors. Experiments for even more learning rates are given in the Appendix. With regard to conventional pricing strategies: this is actually an existing point of failure in existing data markets. Going on any existing data markets quick reveals the fact that current data pricing are largely random: for example, IMDb Essential Metadata for Movies is listed for 150,000 USD for its monthly subscription, and this number comes out arbitrary, without providing insight into how or why it's listed at this price. This is precisely our reason for devising the instrumental value scheme, as it gives a solid pricing framework that people can analyze or build up on, rather than the conventional ad-hoc, intrinsic value-based pricing that is hard to reason about or compare with.
>
> We hope the above explanations give more insight into the design rationale and merits of our paper.

---

> > ### Author Rebuttal · Reviewer_YR6p · 2026-04-02
> >
> > I appreciate the authors’ reply. My question has been resolved. Overall, I tend to maintain my score.

---

> > > ### Author Response · Authors · 2026-04-02
> > >
> > > We thank the reviewer for taking the time and effort to read our reply, and we are glad that our response helped clarify any lingering concerns.

---

### Official Review · Reviewer_nocx · 2026-03-15

**Soundness:** 3
**Presentation:** 2
**Significance:** 3
**Originality:** 3
**Overall Recommendation:** 4
**Confidence:** 3

**Summary:**

This paper studies pricing in a data-augmented AutoML marketplace. Rather than pricing raw datasets directly, the paper proposes pricing the instrumental value of a dataset through the quality improvement of the resulting trained model beyond what can be achieved using the buyer’s original training data alone. In the proposed market, a buyer submits a task and training data, the platform searches over additional data/model combinations, reveals a sequence of discovered model-quality states over time, and charges according to a pricing curve over model-performance levels. The buyer is modeled as solving an optimal stopping problem over this revealed trajectory, where the quality dynamics are represented as a Markov chain. The paper then proposes an MILP-based method for approximately optimal pricing from sampled trajectories, a method for learning the prior over buyer types, and an augmentation/model-discovery component for evaluation.

**Compliance With Llm Reviewing Policy:**

Affirmed.

**Final Justification:**

I share Reviewer jud7’s view that the current manuscript requires substantial revision, particularly with regard to the modeling justification for the buyer’s reasoning, the use of the pricing curve, and the extent to which the framework generalizes beyond the present modeling assumptions. I hope these issues will be adequately addressed in the final version.

**Key Questions For Authors:**

Please refer to the weaknesses above.

**Limitations:**

yes.

**Strengths And Weaknesses:**

## Strengths
The paper raises an interesting idea: pricing datasets in the marketplace according to their marginal contribution to downstream model performance, rather than according to raw dataset characteristics or training cost. This perspective is appealing and more meaningful than directly charging based on training time. The paper also attempts to provide a relatively comprehensive treatment of the framework, including buyer-side optimization, pricing under known parameters, and learning when parameters are unknown.


## Weaknesses
Overall, I found the paper hard to follow, and I was not convinced by several of the modeling choices.

First, some of the terminology is confusing, especially given that the overall model is already fairly complicated. For example, “additional features” appears to refer to external datasets sold in the marketplace, while “metric” is used to denote a model-quality level under some evaluation measure rather than the metric itself. This makes the paper harder to parse than necessary.

Second, the discovery procedure appears to search jointly over both additional datasets and models. If so, the models discovered at times $t$ and $t+1$ may come from very different training pipelines or augmentation choices. In that case, it is not clear whether modeling the process as a Markovian state evolution over model quality alone is the right abstraction. It is also not clear whether the implied search process is computationally realistic, since the platform may effectively need to train many different candidate models.

Third, I am not convinced that optimal stopping is the right model for buyer behavior. If the buyer is assumed to be sophisticated enough to solve the induced stopping problem, then it also seems natural to consider an alternative formulation in which the buyer simply specifies target quality level, budget, or performance-cost tradeoff and asks the platform to search for a model meeting that target.

Finally, it is not clear to me why the paper repeatedly emphasizes the novelty of incorporating the buyer’s original training data. Taken together, although the paper emphasizes the practicality of the framework, I found the overall model too complicated and several of its key modeling choices in need of stronger justification.

Regarding the market design itself, it may also be valuable to analyze the welfare or utility of the sellers who provide the datasets/models in the marketplace, especially if the paper aims to study a broader marketplace mechanism rather than only the buyer-platform interaction.

## Other Comments
- left line 140: a $\mathbb{R}$ seems to be missing in $v^\theta(\cdot) \in_{+}^Q$
- when introducing the stopping time in left line 172, it may be better to clarify that it is endogenously decided rather than given
- some notations seems missing in left line 267 and 268 : $X^{(t)}$ for some $X$, instead of jsut $^{(t)}$
- right line 259: Is $\operatorname{Pr}\left(q, q_{t-1}, z_t\right)=\operatorname{Pr}_t\left(q \mid q_{t-1}\right) \cdot \mathbb{1}_{q>z_t}$ actually $\operatorname{Pr}\left(q, q_{t-1}, z_t\right)=\operatorname{Pr}_t\left(q \mid q_{t-1}\right) \cdot \mathbb{1}_{q\leq z_t}$?
- rigt line 275: $\mathbf{P}$ is not introduced.
- right line 287: "train model" -> "train models"

---

> ### Author Rebuttal · Authors · 2026-03-31
>
> We thank the reviewer for the great feedback. We will clarify terminologies, provide more justification to our modeling. We hope these will demonstrate the value of our work more clearly.
>
> 1. meaning of "additional features”
>
> Thanks for pointing this out. We should have elaborated on this more in the paper. The reason we are calling these "additional features" when referring to external datasets is due to how we use the data: primarily through the JOIN operator that augments the starting dataset with more features (horizontal augmentation). We are happy to rename this if it causes more confusion than clarity.
>
> 2. meaning of "metric"
>
> The reviewer is correct that we are overloading the term "metric" in the paper, which refers to the realized model quality level. The reason is that the buyers are bringing how they want the "goodness of the model" to be measured themselves, so different buyers can use different metrics. We are happy to rename it.
>
> 3. Comparing optimal stopping time with alternative formulations
>
> Certainly, the suggested alternative formulation where buyers bring in a target quality level or budget and search for a qualified model can work. We note that our formulation not only can support such configuration easily, but offers the buyer more. With minimal to no modifications, we can extend to the suggested formulation by keep running the discovery until the budget is exhausted, then search over all discovered models and select one that meets buyers' needs. Yet, our formulation gives buyers the additional flexibility, lower participation risk, and transparency.
>
> Flexibility: Our design lets buyers adapt online to the realized search trajectory, instead of forcing them to commit ex ante to a fixed target quality or budget rule. This is valuable because the usefulness of external data is inherently uncertain before discovery begins, so an adaptive stopping rule is more expressive than a fixed target in many cases: buyers can react to the performance metric evolution in real time, purchase, or leave at any time.
>
> Lower risk and transparency: Under a pure budget/target-driven interface, buyers need to commit to a specific configuration without seeing how promising/likely the search is. In our market, buyers may stop early if discovered trajectories differ from their initial anticipations. This reduces the risk of spending much search cost for little value. Our menu-based design may also contain multiple attractive options, which buyers only learn after seeing the menu. An ex-ante target-quality/cost design loses this richness.
>
> Furthermore, buyers don't even need to solve the DP exactly: even less sophisticated buyers can observe the trajectory evolution, and pick one from the menu that best fits their needs, and they will already benefit from the richness of the menu. Thus, we believe our interaction protocol provides practical benefits.
>
> 4. Computational Feasibility of the Discovery Engine
>
> We have done extensive experiments to verify this. For empirical data points: over a repository with significant size (70K datasets with 30M columns and 3B rows), it reaches convergence between 30 and 50 minutes, on average for 1000 different regression and classification tasks. This illustrates the discovery engine's practicality, which is the dominant cost of our market design.
>
> 5. The Markovian Assumption
>
> Excellent question. We agree that the underlying discovery procedure searches jointly over augmentations and models, so the latent state is richer than model quality alone. Our Markov assumption is better understood as a reduced-form approximation on the observable model quality trajectory, rather than as a literal claim that the full hidden search state is one-dimensional. This approximation is motivated by modern optimization-driven training, where future performance is often largely determined by current performance, and operationally it is learned from empirical metric trajectories generated by the discovery engine. Practically, the performance of our pricing engine is evaluated in RQ2, and we can indeed capture more revenue with our pricing scheme.
>
> We will clarify this in the revised version.
>
> 6. Seller Welfare
>
> This is an important question to answer. Since we are already addressing many other aspects in this work, from design, to discovery, to pricing, to prior learning, we think seller analysis is out of the scope of this particular paper, but is a great future direction.
>
> 7. "emphasis on incorporating the buyer’s original training data"
>
> We mostly incorporate this to contrast against existing data market's interaction protocol, where pricing is arbitrary and buyers have no idea on usefulness of a dataset to be purchased. With buyers bringing their own datasets and observing the instrumental value, they face much lower participation risk. We are happy to de-emphasize this if this point is clear.
>
> We hope the above explanations give more insight into the design rationale and merits of our paper.

---

> > ### Author Rebuttal · Reviewer_nocx · 2026-04-03
> >
> > Thank you to the authors for their detailed responses to my questions. I share Reviewer jud7’s view that the current manuscript requires substantial revision, particularly with regard to the modeling justification for the buyer’s reasoning, the use of the pricing curve, and the extent to which the framework generalizes beyond the present modeling assumptions. I hope these issues will be adequately addressed in the final version.

---

> > > ### Author Response · Authors · 2026-04-04
> > >
> > > Thank you for acknowledging our rebuttal. We are confident that we will be able to make space for the required edits in the updated version of the paper, since those do not need much additional work, and do not concern the core of our technical framework and foundation. With our response hopefully clarifying the existing questions the reviewer may have, we are wondering if that is sufficient to mount to a reassessment of the merit of the paper, and whether the reviewer now think that the strengths of our paper can outweigh its weaknesses.

---

### Decision · Program_Chairs · 2026-04-30

**Decision:**

Accept (regular)

**Comment:**

Scores settle at 4/5/4/4, with all reviewers positive after rebuttal.

The paper addresses a practical and underexplored problem, pricing data in AutoML marketplaces based on instrumental value rather than raw dataset characteristics, with a relatively complete framework spanning discovery, buyer-side optimal stopping, MILP-based pricing, and prior learning. The rebuttal provided concrete scalability evidence and justified key modeling decisions including the Markovian approximation and the optimal stopping formulation.

The paper requires substantial revision before camera-ready, as jud7 and nocx both note. Confusing terminology ("metric" overloaded to mean realized model quality, "additional features" for external datasets) and notation inconsistencies (the relationship between q and t) need to be cleaned up throughout. A dedicated limitations section is missing and must be added. The experimental scope is limited to a single data market with simulated buyer behavior, which weakens generalizability claims.

These are presentation and scope issues rather than fundamental flaws. The core contribution is sound and merits publication with the promised revisions.